# MiniLLM: Knowledge Distillation of Large Language Models

**Yuxian Gu**[1,2][*], **Li Dong**[2], **Furu Wei**[2], **Minlie Huang**[1][†]
[1]The CoAI Group, Tsinghua University  [2]Microsoft Research
guyx21@mails.tsinghua.edu.cn  {lidong1,fuwei}@microsoft.com
aihuang@tsinghua.edu.cn

## Abstract

Knowledge Distillation (KD) is a promising technique for reducing the high computational demand of large language models (LLMs). However, previous KD methods are primarily applied to white-box classification models or training small models to imitate black-box model APIs like ChatGPT. How to effectively distill the knowledge of white-box LLMs into small models is still under-explored, which becomes more important with the prosperity of open-source LLMs. In this work, we propose a KD approach that distills LLMs into smaller language models. We first replace the *forward* Kullback-Leibler divergence (KLD) objective in the standard KD approaches with *reverse* KLD, which is more suitable for KD on generative language models, to prevent the student model from overestimating the low-probability regions of the teacher distribution. Then, we derive an effective optimization approach to learn this objective. The student models are named **MiniLLM**. Extensive experiments in the instruction-following setting show that MiniLLM generates more precise responses with higher overall quality, lower exposure bias, better calibration, and higher long-text generation performance than the baselines. Our method is scalable for different model families with 120M to 13B parameters. Our code, data, and model checkpoints can be found in https://github.com/microsoft/LMOps/tree/main/minillm.

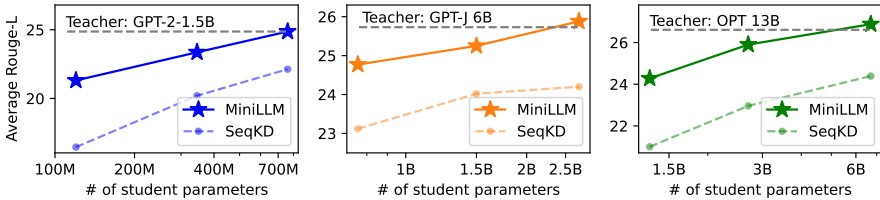

Figure 1: The comparison of MiniLLM with the sequence-level KD (SeqKD)[1] in terms of the average Rouge-L score on the evaluation sets. **Left**: GPT-2-1.5B as the teacher and GPT-2 125M, 340M, 760M as the students. **Middle**: GPT-J 6B as the teacher and GPT-2 760M, 1.5B, GPT-*Neo* 2.7B as the students. **Right**: OPT 13B as the teacher and OPT 1.3B, 2.7B, 6.7B as the students.

## 1 Introduction

With the rapid development of large language models (LLMs; Brown et al., 2020; Han et al., 2021; Bommasani et al., 2021; Chowdhery et al., 2022; OpenAI, 2023), a common technique to reduce their high computational resource demand is knowledge distillation (KD; Hinton et al., 2015), where we train a small student model with supervision from a large teacher model. Two categories of KD are commonly applied: *black-box* KD, where only the teacher-generated texts are accessible, and *white-box* KD, where the teacher model's output distribution or intermediate hidden states are also available (Jianping et al., 2021). Recently, *black-box* KD has shown promising results in fine-tuning

---

[*]Contribution during an internship at Microsoft Research.

[†]Corresponding author.

[1]SeqKD (Kim & Rush, 2016; Chiang et al., 2023; Taori et al., 2023) directly trains the student model on the texts generated by the teacher model, which is a widely used KD method for language models.

small models on the prompt-response pairs generated by LLM APIs (Taori et al., 2023; Chiang et al., 2023; Wu et al., 2023; Peng et al., 2023). With the emergence of more open-source LLMs (Zhang et al., 2022; Touvron et al., 2023), *white-box* KD becomes more valuable for both research communities and industry sectors because student models receive better signals from the output distribution and hidden states of teacher models, thereby potentially resulting in higher performance. However, *white-box* KD approaches are mostly studied for small ($< 1B$ parameters) language understanding models (Sanh et al., 2019; Wang et al., 2020), while *white-box* KD for LLMs is yet to be explored.

In this work, we investigate *white-box* KD of LLMs where the output distribution of the teacher model is available. We argue that the standard KD objectives (Kim & Rush, 2016; Song et al., 2020; Chiang et al., 2023; Taori et al., 2023) are sub-optimal for LLMs that perform tasks in a generative manner. Given the teacher distribution $p(\boldsymbol{y}|\boldsymbol{x})$ and the student distribution $q_\theta(\boldsymbol{y}|\boldsymbol{x})$ parameterized by $\theta$, standard KD objectives (including several variants for sequence-level models) essentially minimize the approximated *forward* Kullback-Leibler divergence (KLD) between the teacher and the student distribution, termed as $\mathrm{KL}[p||q_\theta]$, which forces $q_\theta$ to cover all modes of $p$. For text classification tasks, $\mathrm{KL}[p||q_\theta]$ works well because the output space usually consists of a finite number of classes such that both $p(\boldsymbol{y}|\boldsymbol{x})$ and $q_\theta(\boldsymbol{y}|\boldsymbol{x})$ have few modes. However, for open-ended text generation tasks, the output spaces are much more complex, and $p(\boldsymbol{y}|\boldsymbol{x})$ can contain many more modes than what $q_\theta(\boldsymbol{y}|\boldsymbol{x})$ can express due to the limited model capacity. Minimizing *forward* KLD causes $q_\theta$ to assign unreasonably high probabilities to the void regions of $p$ (Malinin & Gales, 2019) and produces very unlikely samples under $p$ during free-run generation (Huszár, 2015).

To alleviate this problem, we propose to minimize *reverse* KLD, $\mathrm{KL}[q_\theta||p]$, widely used in computer vision (Lee et al., 2023) and reinforcement learning (Czarnecki et al., 2019). Compared to $\mathrm{KL}[p||q_\theta]$, minimizing $\mathrm{KL}[q_\theta||p]$ causes $q_\theta$ to seek the major modes of $p$, and assign low probabilities to $p$'s void regions (Minka et al., 2005), as illustrated in Table 2 and discussed in Section 2.1. In LLM text generation, this means that the student model avoids learning too many long-tail variants (Holtzman et al., 2020) in the teacher model's distribution and focuses on the correctness of

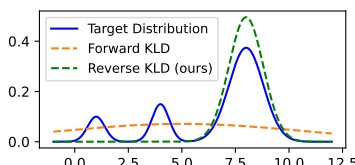

Figure 2: We fit a Gaussian mixture distribution with a single Gaussian distribution using *forward* KLD and *reverse* KLD.

the generated cotents, which is critical in practical scenarios that require truthfulness and reliability (Ji et al., 2023b). To optimize $\min_\theta \mathrm{KL}[q_\theta||p]$, as shown in Section 2.2, we derive the gradient of the objective with Policy Gradient (Sutton et al., 1999). To further stabilize and accelerate training, we propose (1) single-step decomposition to reduce variance, (2) teacher-mixed sampling to alleviate reward hacking, and (3) length normalization to eliminate the length bias. Finally, we introduce the overall KD algorithm in Section 2.3. Our student models are named **MINILLM**, indicating our method is suitable and works well for compressing large (generative) language models.

We apply our method to various generative language models (Radford et al., 2019; Zhang et al., 2022; Touvron et al., 2023) with sizes ranging from 120M to 13B in the instruction-following setting (Sanh et al., 2022; Wei et al., 2022a) that covers a large range of NLP tasks. We use 5 datasets with Rouge-L (Lin, 2004), human judgment, and the GPT-4 feedback for evaluation. Experiments show that MINILLM consistently outperforms standard KD baselines on all the datasets and scales up well from 120M to 13B models (see Figure 1). More analysis shows that MINILLM yields lower exposure bias, better calibration, and higher long response generation performance.

## 2 METHOD

We consider conditional text generation where the model produces a response $\boldsymbol{y} = \{y_t\}_{t=1}^T$ conditioning on a prompt $\boldsymbol{x}$ sampled from the distribution $p_{\boldsymbol{x}}$, which is typically how LLMs perform tasks. We formulate KD as an optimization problem to minimize the difference between a fixed teacher model distribution $p(\boldsymbol{y}|\boldsymbol{x})$ and a student model distribution $q_\theta(\boldsymbol{y}|\boldsymbol{x})$ parameterized by $\theta$. The standard KD methods approximately[2] minimize the *forward* KLD: $\mathrm{KL}[p||q_\theta] = \mathbb{E}_{\boldsymbol{x}\sim p_{\boldsymbol{x}}, \boldsymbol{y}\sim p'} \log \frac{p(\boldsymbol{y}|\boldsymbol{x})}{q_\theta(\boldsymbol{y}|\boldsymbol{x})}$, where $p'$ can be real data distribution (word-level KD) or teacher distribution $p$ (sequence-level KD).

---

[2]We say "approximately" because for word-level KD, $\boldsymbol{y}$ is sampled from the real distribution, not the teacher distribution. For a strong enough teacher model, we can consider the two distributions approximately the same.

Figure 3: Comparison between sequence-level KD (left) and MINILLM (right). Sequence-level KD forces the student to memorize all samples generated by the teacher model, while MINILLM improves its generated texts with the teacher model's feedback.

Though widely used, $\text{KL}[p||q_\theta]$ tends to overestimate the void regions of $p$ in text generation tasks when $q_\theta$ is insufficiently expressive (Ji et al., 2023a). KD for LLMs fits the case because LLMs perform tasks in a generative manner, such that the low-capacity student models cannot perfectly imitate the complex text generation distribution of the teacher models or humans.

## 2.1 MINILLM: KNOWLEDGE DISTILLATION WITH *Reverse* KLD

We consider minimizing the *reverse* KLD between the student and teacher model distributions as the learning objective for MINILLM:

$$
\theta = \arg\min_\theta \mathcal{L}(\theta) = \arg\min_\theta \text{KL}[q_\theta||p]
$$
$$
= \arg\min_\theta \left[ - \mathbb{E}_{\boldsymbol{x}\sim p_{\boldsymbol{x}}, \boldsymbol{y}\sim q_\theta} \log \frac{p(\boldsymbol{y}|\boldsymbol{x})}{q_\theta(\boldsymbol{y}|\boldsymbol{x})} \right]. \tag{1}
$$

Minimizing *reverse* KLD has been shown to cause the mode-seeking behavior in generative modeling (Huszár, 2015; Nowozin et al., 2016; Chen et al., 2018; Lee et al., 2023), where $q_\theta$ assigns high probabilities to $p$'s large modes and ignore the small ones (illustrated in a toy experiment in Figure 2). In this work, we first study this property for KD of LLMs in text generation. Minimizing *forward* KLD causes $q_\theta$ to place large probability mass on the zero-probability places of $p$, corresponding to the low-quality text generation in practice, while *reverse* KLD focuses on $p$'s major modes, which is crucial to ensure the correctness and faithfulness of text generation. As illustrated in Figure 3, unlike sequence-level KD, MINILLM that minimizes *reverse* KLD does not force $q_\theta$ to fit all $\boldsymbol{y}$ sampled from the teacher distribution $p$. Instead, it encourages the student to generate samples preferred by the teacher within its own capacities, which is more possible to achieve. Interestingly, we also find another perspective of understanding MINILLM motivated by Inverse Reinforcement Learning (Ziebart et al., 2008). We present the related derivation in Appendix A.1.

## 2.2 OPTIMIZATION WITH POLICY GRADIENT

**Gradient Derivation** We notice that the gradient of the objective function $\mathcal{L}(\theta)$ in Equation 1 can be derived using the Policy Gradient Theorem (Williams, 1992; Haarnoja et al., 2017):

$$
\nabla \mathcal{L}(\theta) = - \mathbb{E}_{\boldsymbol{x}\sim p_{\boldsymbol{x}}, \boldsymbol{y}\sim q_\theta(\cdot|\boldsymbol{x})} \sum_{t=1}^{T} (R_t - 1) \nabla \log q_\theta(y_t|\boldsymbol{y}_{<t}, \boldsymbol{x}), \tag{2}
$$

where $T = |\boldsymbol{y}|$ and $R_t = \sum_{t'=t}^{T} \log \frac{p(y_{t'}|\boldsymbol{y}_{<t'}, \boldsymbol{x})}{q_\theta(y_{t'}|\boldsymbol{y}_{<t'}, \boldsymbol{x})}$ is the accumulation of $r_{t'} = \log \frac{p(y_{t'}|\boldsymbol{y}_{<t'}, \boldsymbol{x})}{q_\theta(y_{t'}|\boldsymbol{y}_{<t'}, \boldsymbol{x})}$ that measures the quality of each step's generation. Intuitively, the generated texts are supposed to have high probabilities under the teacher distribution by increasing $p(y_{t'}|\boldsymbol{y}_{<t'}, \boldsymbol{x})$, but simultaneously stay diverse by lowering $q_\theta(y_{t'}|\boldsymbol{y}_{<t'}, \boldsymbol{x})$. The expectation in Eq. 2 is computed by Monte-Carlo sampling. Full derivation can be found in Appendix A.2. However, policy gradient suffers from high variance and reward hacking (Skalse et al., 2022), despite some subsequent solutions (Schulman et al., 2017). Besides, we notice that $R_t$ favors short sentences, which causes the student model to output empty responses. Therefore, we propose three strategies to mitigate these problems.

**Single-Step Decomposition** Czarnecki et al. (2019) has found that the single-step generation quality $r_t$ is critical to the training variance because the error in the front tokens accumulates along the whole sentence. To pay more attention to $r_t$, we re-write $\nabla\mathcal{L}(\theta)$ to decompose $r_t$ from $R_t$ and

directly compute the gradient of $\mathbb{E}_{y_t \sim q_\theta(t)}[r_t]$ (see Appendix A.3 for the full derivation):

$$\nabla \mathcal{L}(\theta) = \underset{\substack{\boldsymbol{x} \sim p_{\boldsymbol{x}} \\ \boldsymbol{y} \sim q_\theta(\cdot|\boldsymbol{x})}}{\mathbb{E}} \left[ -\sum_{t=1}^{T} \nabla \underset{y_t \sim q_\theta(t)}{\mathbb{E}} [r_t] \right] + \underset{\substack{\boldsymbol{x} \sim p_{\boldsymbol{x}} \\ \boldsymbol{y} \sim q_\theta(\cdot|\boldsymbol{x})}}{\mathbb{E}} \left[ -\sum_{t=1}^{T} R_{t+1} \nabla \log q_\theta(y_t|\boldsymbol{y}_{<t}, \boldsymbol{x}) \right] \quad (3)$$
$$= (\nabla \mathcal{L})_{\text{Single}} + (\nabla \mathcal{L})_{\text{Long}},$$

where $q_\theta(t) = q_\theta(\cdot|\boldsymbol{y}_{<t}, \boldsymbol{x})$. Note that $\mathbb{E}_{y_t \sim q_\theta(t)}[r_t]$ can be computed directly by summing over the vocabulary instead of using Monte-Carlo sampling and is derivable with respect to $\theta$. This decomposition gives a more precise and efficient estimation of the single-step generation quality, which reduces the variance during training and accelerates convergence.

**Teacher-Mixed Sampling** We observe reward hacking (Skalse et al., 2022) when training with Eq. 2 because $q_\theta$ sometimes produces degenerated sentences $\boldsymbol{y}$ that receive high scores from the teacher (e.g., repeated phrases) during sampling, especially for small student models. To create a better sampling distribution, we mix the teacher and the student distribution at each time step:

$$\widetilde{p}(y_t|\boldsymbol{y}_{<t}, \boldsymbol{x}) = \alpha \cdot p(y_t|\boldsymbol{y}_{<t}, \boldsymbol{x}) + (1 - \alpha) \cdot q_\theta(y_t|\boldsymbol{y}_{<t}, \boldsymbol{x}), \quad (4)$$

where $\alpha$ controls the strength of the teacher mix-in. Sampling from $\widetilde{p}$ suppresses low-quality generation with the teacher's help and alleviates reward hacking. We re-write $(\nabla \mathcal{L})_{\text{Single}}$ and $(\nabla \mathcal{L})_{\text{Long}}$ with importance sampling to get to an unbiased estimator of the gradient (Precup et al., 2000):

$$(\nabla \mathcal{L})_{\text{Single}} = - \underset{\boldsymbol{x} \sim p_{\boldsymbol{x}}, \boldsymbol{y} \sim \widetilde{p}(\cdot|\boldsymbol{x})}{\mathbb{E}} \left[ \sum_{t=1}^{T} w_t \nabla \underset{y_t \sim q_\theta(t)}{\mathbb{E}} [r_t] \right],$$
$$(\nabla \mathcal{L})_{\text{Long}} = - \underset{\boldsymbol{x} \sim p_{\boldsymbol{x}}, \boldsymbol{y} \sim \widetilde{p}(\cdot|\boldsymbol{x})}{\mathbb{E}} \left[ \sum_{t=1}^{T} w_t R_{t+1} \nabla \log q_\theta(y_t|\boldsymbol{y}_{<t}, \boldsymbol{x}) \right], \quad (5)$$

where $w_t = \prod_{t'=1}^{t} \frac{q_\theta(y_{t'}|\boldsymbol{y}_{<t'}, \boldsymbol{x})}{\widetilde{p}(y_{t'}|\boldsymbol{y}_{<t'}, \boldsymbol{x})}$ is the importance weight. However, $w_t$ brings high variance in practice because it requires multiplying per-token importance weight over multiple time steps, and thus the variance of each step accumulates. Therefore, we approximately set $w_t \approx \frac{q_\theta(y_t|\boldsymbol{y}_{<t}, \boldsymbol{x})}{\widetilde{p}(y_t|\boldsymbol{y}_{<t}, \boldsymbol{x})}$ to reduce the variance of the estimator in Eq. 5 (Serban et al., 2017; Levine et al., 2020).

**Length Normalization** We found that long sequences tend to have small $R_{t+1}$, which encourages the model to produce short responses. Therefore, we add length normalization to $R_{t+1}$ in Eq. 3:

$$R_{t+1}^{\text{Norm}} = \frac{1}{T - t - 1} \sum_{t'=t+1}^{T} \log \frac{p(y_{t'}|\boldsymbol{y}_{<t'}, \boldsymbol{x})}{q_\theta(y_{t'}|\boldsymbol{y}_{<t'}, \boldsymbol{x})}. \quad (6)$$

**In Summary** Combining the strategies listed above, we have the final optimization gradient:

$$\nabla \mathcal{L}(\theta) = - \underset{\substack{\boldsymbol{x} \sim p_{\boldsymbol{x}} \\ \boldsymbol{y} \sim \widetilde{p}(\cdot|\boldsymbol{x})}}{\mathbb{E}} \left[ \sum_{t=1}^{T} w_t \left[ \underbrace{\nabla \sum_{y' \in V} q_\theta(y'|\boldsymbol{y}_{<t}, \boldsymbol{x}) \log \frac{p(y'|\boldsymbol{y}_{<t}, \boldsymbol{x})}{q_\theta(y'|\boldsymbol{y}_{<t}, \boldsymbol{x})}}_{(\nabla \mathcal{L})_{\text{Single}} \text{ part}} + \underbrace{R_{t+1}^{\text{Norm}} \frac{\nabla q_\theta(y_t|\boldsymbol{y}_{<t}, \boldsymbol{x})}{q_\theta(y_t|\boldsymbol{y}_{<t}, \boldsymbol{x})}}_{(\nabla \mathcal{L})_{\text{Long}}^{\text{Norm}} \text{ part}} \right] \right], \quad (7)$$

where $V$ is the vocabulary size of the language model and $(\nabla \mathcal{L})_{\text{Long}}^{\text{Norm}}$ is $(\nabla \mathcal{L})_{\text{Long}}$ with $R_{t+1}^{\text{Norm}}$.

## 2.3 TRAINING ALGORITHM

We start from a student model pre-trained on a large long-document corpus $\mathcal{D}_{\text{PT}}$. The algorithm to train MINILLM adapts the student model to a text generation task with dataset $\mathcal{D}$. We assume that there is a teacher model performing well on $\mathcal{D}$, such as an LLM fine-tuned on $\mathcal{D}$ (Taori et al., 2023; Chiang et al., 2023) or that with good task-generalization (Chung et al., 2022; OpenAI, 2023).

In the training algorithm, we first fine-tune the student model on $\mathcal{D}$ and pick the checkpoint with the lowest loss as an initialization for the following training. Then, we compute the gradients $(\nabla \mathcal{L})_{\text{Single}}$ and $(\nabla \mathcal{L})_{\text{Long}}^{\text{Norm}}$ based on Eq. 5 and Eq. 6, with a clipping strategy (Schulman et al., 2017) added to further improve stability. Same as Ouyang et al. (2022), we include a language modeling loss $\mathcal{L}_{\text{PT}} = -\mathbb{E}_{\boldsymbol{d} \sim \mathcal{D}_{\text{PT}}} \log q_\theta(\boldsymbol{d})$ to preserve the model performance on canonical NLP benchmarks. The student model is finally updated using a combination of gradients $(\nabla \mathcal{L})_{\text{Single}} + (\nabla \mathcal{L})_{\text{Long}}^{\text{Norm}} + \nabla \mathcal{L}_{\text{PT}}$. The whole training pipeline is similar to Reinforcement Learning from Human Feedback (RLHF; Ouyang et al., 2022). We present the details of the MINILLM training algorithm in Appendix B.

## 3 EXPERIMENTS

### 3.1 EXPERIMENTAL SETUP

We take instruction-following (Ouyang et al., 2022) as the conditional text generation task, where models are trained to generate responses according to the instructions. We fine-tune a large model on the dataset $\mathcal{D}$ consisting of instruction-response pairs as the teacher model. Then, we compare different KD methods on $\mathcal{D}$ by evaluating the student model's instruction-following performance.

**Base Models** Our student models come from three model families with various sizes: GPT-2 (Radford et al., 2019) (120M, 340M, 760M), OPT (Zhang et al., 2022) (1.3B, 2.7B, 6.7B), and LLaMA (Touvron et al., 2023) (7B). For teacher models of each model family, we use GPT-2-1.5B, OPT-13B, and LLaMA-13B respectively. These models are fine-tuned on $\mathcal{D}$ in advance. We also present the results using GPT-J (Wang & Komatsuzaki, 2021) as the teacher in Appendix D.2.

**Training** We construct the training data from `databricks-dolly-15K`[3] consisting of 15K human-written instruction-response pairs. We filter out samples that exceed the context length of the models. Then, we randomly split 0.5K and 1K samples for validation and testing, respectively, leaving about 12.5K examples for training. For $\mathcal{D}_{\mathrm{PT}}$, we use OpenWebText (Gokaslan et al., 2019) for the GPT-2 family and the RoBERTa training corpus (Liu et al., 2019) for other models. We set the teacher-mix-in strength $\alpha = 0.2$ throughout the experiments in Eq. 4. We use Rouge-L (Lin, 2004) scores on the validation set to search for hyper-parameters because it aligns better with human preference than validation losses (Wang et al., 2022). More details are shown in Appendix C.1.

**Evaluation** We evaluate the trained models on 5 instruction-following datasets:

- **Dolly**: the 500-sample test set we split from the `databricks-dolly-15K` dataset.
- **SelfInst** (Wang et al., 2023): A user-oriented instruction-following set with 252 samples.
- **Vicuna** (Chiang et al., 2023): The 80 challenging questions used in the Vicuna evaluation.
- **S-NI**: The test set of SUPER-NATURALINSTRUCTIONS (Wang et al., 2022) consisting of 9K samples ranging from 119 tasks. Following Peng et al. (2023), we split the set into 3 subsets whose ground truth response lengths lie in $[0, 5]$, $[6, 10]$, and $[11, +\infty]$. We use the $[11, +\infty]$ subset in Section 3.2 and conduct an analysis on all subsets in Section 3.3.
- **UnNI**: The core set of UNNATURALINSTRUCTIONS (Honovich et al., 2023) containing 60K samples. Similar to S-NI, we first conduct the evaluations on the randomly sampled 10K examples in the $[11, +\infty]$ subset, followed by an analysis of the performance on all subsets in Appendix D.4.

We adopt two metrics to evaluate the model-generated responses:

- **R-L**: The Rouge-L (Lin, 2004) score to measure the precision of the model generation. Wang et al. (2022) has shown that Rouge-L is suitable for large-scale instruction-following evaluation.
- **HumanEval**: We conduct human evaluations on the SelfInst dataset following Peng et al. (2023) by asking volunteers to compare two responses produced by different models and annotate "Win", "Tie", or "Loss". More human evaluation details can be found in Appendix C.3.

For all test sets, we sample the responses with the temperature = 1 and report the average scores of 5 generations for each prompt with different random seeds. We also include a supplemental evaluation using the GPT-4 feedback (Zheng et al., 2023) by asking GPT-4 (OpenAI, 2023) to compare model-generated responses with the ground truth answers and present the results in Appendix D.1.

**Baselines** We consider three baselines in our main experiment:

- **SFT w/o KD** directly fine-tunes the student model on $\mathcal{D}$ supervised by the golden responses.
- **KD** (Sanh et al., 2019; Song et al., 2020) fine-tunes the student model on $\mathcal{D}$ using the teacher distribution as the supervision at each token step, also known as word-level KD.
- **SeqKD** (Kim & Rush, 2016; Chiang et al., 2023; Taori et al., 2023) fine-tunes the student model on the data generated by the teacher model.

---

[3] https://github.com/databrickslabs/dolly/tree/master

| Model | #Params | Method | Dolly | SelfInst | Vicuna | S-NI | UnNI |
|-------|---------|--------|-------|----------|--------|------|------|
| | 1.5B | Teacher | 27.6 | 14.3 | 16.3 | 27.6 | 31.8 |
| GPT-2 | 120M | SFT w/o KD | 23.3 | 10.0 | 14.7 | 16.3 | 18.5 |
| | | KD | 22.8 | 10.8 | 13.4 | 19.7 | 22.0 |
| | | SeqKD | 22.7 | 10.1 | 14.3 | 16.4 | 18.8 |
| | | MINILLM | **24.6** | **13.2** | **16.9*** | **25.3** | **26.6** |
| | 340M | SFT w/o KD | **25.5** | 13.0 | 16.0 | 25.1 | 28.1 |
| | | KD | 25.0 | 12.0 | 15.4 | 23.7 | 24.6 |
| | | SeqKD | 25.3 | 12.6 | 16.9* | 22.9 | 23.3 |
| | | MINILLM | 25.4 | **15.6*** | **17.7*** | **27.4** | **30.8** |
| | 760M | SFT w/o KD | 25.4 | 12.4 | 16.1 | 21.5 | 24.0 |
| | | KD | 25.9 | 13.4 | 16.9* | 25.3 | 28.0 |
| | | SeqKD | 25.6 | 14.0 | 15.9 | 26.1 | 29.1 |
| | | MINILLM | **26.4** | **15.9*** | **18.3*** | **29.3*** | **34.5*** |
| | 13B | Teacher | 29.2 | 18.4 | 17.8 | 30.4 | 36.1 |
| OPT | 1.3B | SFT w/o KD | 26.0 | 11.4 | 15.6 | 23.1 | 28.4 |
| | | KD | 25.4 | 12.2 | 14.9 | 21.9 | 27.0 |
| | | SeqKD | 26.1 | 12.7 | 16.6 | 21.4 | 28.2 |
| | | MINILLM | **26.7** | **14.8** | **17.9*** | **28.6** | **33.4** |
| | 2.7B | SFT w/o KD | 27.1 | 13.9 | 16.6 | 24.9 | 32.3 |
| | | KD | 25.9 | 13.8 | 16.7 | 26.3 | 30.2 |
| | | SeqKD | 27.5 | 13.3 | 16.5 | 25.3 | 32.3 |
| | | MINILLM | 27.4 | **17.2** | **19.1*** | **30.7*** | **35.1** |
| | 6.7B | SFT w/o KD | 27.6 | 16.4 | 17.8 | 30.3 | 28.6 |
| | | KD | 28.3 | 17.0 | 17.5 | 30.7* | 26.7 |
| | | SeqKD | 28.5 | 17.0 | 17.9* | 30.4 | 28.2 |
| | | MINILLM | **29.0** | **17.5** | **18.7*** | **32.5*** | **36.7*** |
| | 13B | Teacher | 29.7 | 23.4 | 19.4 | 35.8 | 38.5 |
| LLaMA | 7B | SFT w/o KD | 26.3 | 20.8 | 17.5 | 32.4 | 35.8 |
| | | KD | 27.4 | 20.2 | 18.4 | 33.7 | 37.9 |
| | | SeqKD | 27.5 | 20.8 | 18.1 | 33.7 | 37.6 |
| | | MINILLM | **29.0** | **23.2** | **20.7*** | **35.5** | **40.2*** |

Table 1: Evaluation results. We report the average R-L scores across 5 random seeds. The best scores of each model size are **boldfaced**, and the scores where the student model outperforms the teacher are marked with *.

## 3.2 RESULTS

We present the Rouge-L evaluation results in Table 1, from which we have three observations.

*First*, by comparing the overall performance of MINILLM with the baselines, we observe that the model distilled by our KD method outperforms the baselines in almost all cases, when trained with different base models and tested on various evaluation sets. This verifies the good generalization and high overall performance of our KD method. We also find that MINILLM generally works much better on datasets other than Dolly compared with the baselines, indicating its good out-of-distribution generalization.

*Second*, the Rouge-L scores show that the MINILLM produces the most precise responses that have high overlaps with the ground-truth responses. In some cases, especially on Vicuna, S-NI, and UnNI, student models reach even higher Rouge-L scores than the teacher models, which matches the observation in Furlanello et al. (2018). We conjecture that the standard teacher-forcing fine-tuning on $\mathcal{D}$ brings training-inference discrepancy to the teacher model, also known as exposure bias (Bengio et al., 2015). On the contrary, MINILLM is optimized with policy optimization methods, which samples responses from student models during training and thus alleviates exposure bias (Pang & He, 2021). We include further analysis on exposure bias in Section 3.3.

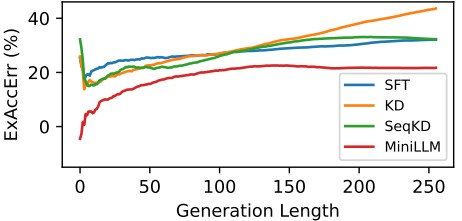

| | SST2 | | BoolQ | |
| | ECE | Acc. | ECE | Acc. |
|---|---|---|---|---|
| Teacher | 0.025 | 93.0 | 0.356 | 74.5 |
| KD | 0.191 | 84.7 | 0.682 | 63.5 |
| SeqKD | 0.243 | 66.5 | 0.681 | 62.8 |
| MINILLM | **0.099** | **89.7** | **0.502** | **67.8** |

Figure 6: The excess error caused by the training-inference discrepancy (ExAccErr) accumulated with the generation length. Lower ExAccErr means the method introduces less exposure bias.

Table 2: The ECE scores and accuracy scores (Acc.) on SST2 and BoolQ datasets. The best scores among student models are **boldfaced**.

*Third*, comparing the results across model sizes and model families, we can see that the improvement of MINILLM is consistent when the base model sizes vary from 120M to 13B across three model families. This tendency is also illustrated in Figure 1, which demonstrates the excellent scalability and generalization of our method in the era of LLMs.

The human evaluation results on the SelfInst dataset based on the LLaMA family are shown in Figure 4. MINILLM obtains better human preference than all the baselines, performing comparably to the teacher model.

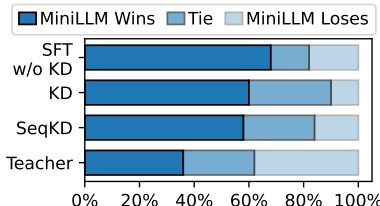

Figure 4: Human evaluation results. We use LLaMA-7B as the student and LLaMA-13B as the teacher.

### 3.3 ANALYSIS

**Scaling Law of Teacher** Although it is intuitive that we can distill better student models from larger teacher models, Mirzadeh et al. (2020) has shown that increasing the teacher models' sizes does not guarantee the improvement of student models, sometimes even harming the distillation performance. It is not clear how MINILLM works when we scale up the teacher models' sizes. Therefore, we compare MINILLM and SeqKD using teacher models with different sizes and fix the size of the student model. We present the results based on the GPT-2 family in Figure 5 and that based on the OPT family in Appendix D.3. We can see that MINILLM constantly outperforms SeqKD, and the student model performance is positively correlated with the teacher model sizes. This shows the potential of our method to compress models with massive parameters.

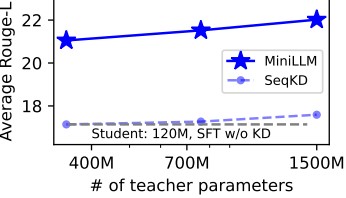

Figure 5: The scaling law of teacher based on the GPT-2 family models. We compare MINILLM and SeqKD with GPT-2-125M as the student and GPT-2 340M, 760M, and 1.5B as teachers.

**Exposure Bias** Language generation models trained to minimize *forward* KLD suffer from exposure bias (Bengio et al., 2015) caused by the discrepancy between teacher-forcing training and free-run generation. When training MINILLM, the student model sees samples generated by itself, alleviating the training-inference mismatch (Pang & He, 2021). In Figure 6, we use the ExAccErr metric (Arora et al., 2022) defined in Appendix C.5 to measure the excess accumulated error due to exposure bias. The experiment is based on GPT-2-125M, with GPT-2-1.5B as the teacher, using Dolly as the test set. For each prompt, we sample 10 responses to reduce the variance. We can see that the ExAccErrs of the baselines continuously grow during generation, while MINILLM has a much lower ExAccErr, and the error stops accumulating in long-text generation ($> 150$ tokens).

**Calibration** OpenAI (2023) has shown that models trained with policy optimization are likely to be poorly calibrated. We test the calibration of MINILLM and the KD baselines on two widely-used text classification datasets: SST2 (Socher et al., 2013) and BoolQ (Clark et al., 2019), based on LLaMA-7B. We design zero-shot classification instructions (see Appendix C.2) and take the

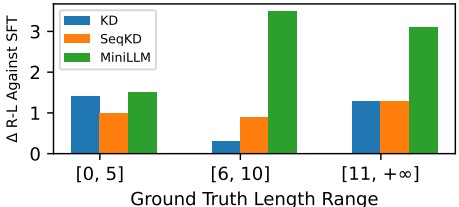

|  | Dolly | | SelfInst | |
|---|---|---|---|---|
|  | Dist-4 | Loss | Dist-4 | Loss |
| Teacher | 99.3 | 3.55 | 99.1 | 4.44 |
| KD | 99.4 | 3.93 | 98.8 | 5.36 |
| SeqKD | 99.3 | 3.91 | 98.8 | 5.22 |
| MINILLM | 99.0 | 3.95 | 98.6 | 5.33 |

Figure 7: The Rouge-L scores of the distilled models against SFT models on the different subsets of S-NI split by the golden responses' length.

Table 3: The distinct 4-grams (Dist-4) and language modeling loss (Loss) on the test sets based on the LLaMA family. MINILLM preserves generation diversity.

probability of the label words to compute the ECE scores (Nixon et al., 2019). From Table 2, we observe that KD and SeqKD models are worse calibrated than the teacher model, which potentially explains their low performance on canonical benchmarks (Gudibande et al., 2023). We suspect that minimizing *forward* KLD causes the models to push high probabilities to void regions of the target distribution, which leads to significant distribution differences between the student and the teacher (see the example in Figure 2). In contrast, MINILLM focuses on accurately learning the major parts of the target distribution and narrows the ECE scores gap between the student and the teacher.

**Performance on Various Response Length** We study the models' performance when the golden response lengths belong to different ranges. In Figure 7, we illustrate the Rouge-L scores of different KD models against the SFT models on three S-NI subsets split by the length of the ground truth responses. We can see that all methods achieve low scores on prompts that expect short responses ($\leq 5$ tokens), probably because most responses in our training set are long sentences, introducing a distribution shift between training and evaluation (Peng et al., 2023). Furthermore, the output spaces of these prompts are relatively small, allowing the student model to cover most modes of the teacher, and thus *reverse* KLD and *forward* KLD have similar performance. For prompts with longer responses ($\geq 6$ tokens), the teacher distribution contains more modes than the students due to the complex output spaces, which shows the advantage of MINILLM against standard KD models. Similar results on UnNI are shown in Appendix D.4.

**Generation Diversity** Caccia et al. (2020) has found that the model optimized by minimizing *reverse* KLD is likely to lose modes, which affects the generation diversity. We follow Pang & He (2021) to discuss generation diversity from three aspects: (i) generating multiple distinct responses given a prompt. (ii) generating linguistically complex responses. (iii) the ability to generate contents that have high coverage of the real data distribution. For (i), we argue that for many NLP applications, generating one **correct** response is sufficient, especially for those scenarios demanding high truthfulness and reliability (Ji et al., 2023b). For (ii) and (iii), we report the responses' distinct 4-gram proportion and the language modeling loss on the test sets in Table 3, using the base models from the LLaMA family (see Appendix C.4 for more details) . We can see that MINILLM preserves the distinct 4-gram proportion in the generated responses and language modeling loss on the test set.

### 3.4 ABLATION STUDIES ON OPTIMIZATION STRATEGIES

We evaluate the effectiveness of the three strategies proposed to stabilize and accelerate optimization in Section 2.2 by distilling a GPT-2-125M model from the GPT-2-1.5B model. More ablation studies can be found in Appendix D.5. In Table 4, we report the best Rouge-L scores on the validation set of each run and the evaluation results of the corresponding checkpoints. We also plot the *reverse* KLD between the student and the teacher during training in Figure 8, where the curves are smoothed by 32 steps. We can see that Teacher-Mixed Sampling and Length Normalization works for stabilizing training. Although the *reverse* KLDs also decrease without these strategies, we find that the models quickly learn to generate repeated, short, or meaningless strings that have high probabilities in the teacher distribution (see examples in Appendix E), which is known as reward hacking (Skalse et al., 2022). This also leads to the low generation performance in Table 4. From Figure 8, we also observe that the Single-Step Decomposition effectively reduces the variance of the training process, which also results in higher scores on the validation and test sets.

|  | Valid.
R-L | Dolly
R-L |
|---|---|---|
| MINILLM | **27.4** | **24.6** |
| w/o Length Norm. | 17.4 | 14.7 |
| w/o Teacher-Mixed | 22.3 | 20.4 |
| w/o Single-Step | 27.0 | 23.7 |

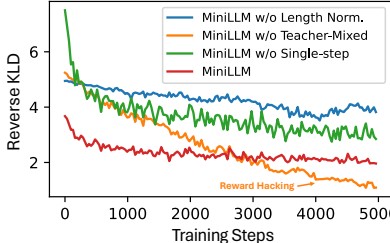

Table 4: The performance on the validation and test set when different combinations of MINILLM optimization strategies are applied.

Figure 8: The *reverse* KLD between the teacher and the students during MINILLM training when different optimization strategies are applied.

## 4 RELATED WORK

**Large Language Models** Large language models (LLMs; Brown et al., 2020; Thoppilan et al., 2022; Chowdhery et al., 2022; OpenAI, 2023; Anil et al., 2023) have shown superior performance by solving various NLP tasks in a generative manner. Recent works apply instruction tuning (Wei et al., 2022a; Sanh et al., 2022; Chung et al., 2022) or learning from human feedback (Ouyang et al., 2022; Bai et al., 2022) to improve the alignment of LLMs with humans further and create general AI assistants (OpenAI, 2022; Google, 2023). There are also efforts to build open-source LLMs (Zhang et al., 2022; Touvron et al., 2023; Biderman et al., 2023) to facilitate research and industry development. Although appealing, the broad capacities of LLMs usually emerge with large model sizes (Kaplan et al., 2020; Wei et al., 2022b) that require massive computational resources. Therefore, model compression is critical for the practical deployment and further research of LLMs.

**Knowledge Distillation** Knowledge distillation (KD; Hinton et al., 2015), as a widely used model compression technique, aims at training a student model with the guidance of a teacher model (Rusu et al., 2015; Sanh et al., 2019; Jianping et al., 2021). In the NLP community, many works apply KD to text classification tasks by mimicking the teacher model's output distribution (Song et al., 2020; Liang et al., 2021; Zhang et al., 2023), hidden states (Jiao et al., 2020; Sun et al., 2019), or attention scores (Wang et al., 2020; 2021). For text generation, the standard KD method is to approximately minimize the *forward* KLD between the student's and the teacher's generation distribution by using the teacher's output at each time step as supervision (Sanh et al., 2019) or direct training on the teacher's generated texts (Kim & Rush, 2016; Taori et al., 2023; Chiang et al., 2023; Peng et al., 2023). In this paper, we minimize the *reverse* KLD, which is more suitable for LLMs when the teacher distribution is available. Concurrent works (Agarwal et al., 2023; Wen et al., 2023) also explore more the distribution discrepancy metrics in KD.

**Distribution Discrepancy Metrics in Text Generation** The distribution discrepancy metrics play a significant role in training text generation models. The *forward* KLD is widely used due to its simplicity when derived as the Maximum Likelihood Estimate (MLE) objective (Zhang & Zhao, 2019). However, previous works show that minimizing *forward* KLD leads to zero-forcing behavior where models try to cover all modes of the target distribution and sacrifice the accuracy of major modes (Huszár, 2015). Some works resort to using other metrics to remedy this problem, such as *reverse* KLD (Jiang et al., 2020), Total Variation Distance (Ji et al., 2023a), and Optimal Transport (Li et al., 2020). Our paper tackles this problem under the scenario of knowledge distillation for LLMs.

## 5 CONCLUSION

In this work, we investigate the problem of distilling the knowledge of LLMs into small language models. We find that the standard distillation methods minimizing the *forward* KLD is sub-optimal in language generation scenarios because the teacher's output distribution contains more modes than the student's, and *forward* KLD forces the student distribution to overestimate the low-probability regions of the teacher distribution. Therefore, we propose MINILLM that minimizes the *reverse* KLD between the teacher and student distribution and design an algorithm to optimize this objective. Extensive experiments show that MINILLM produce more precise responses that have higher overall quality than standard KD models. We also find that MINILLM has lower exposure bias, better calibration, and higher performance in long-text generation with good generation diversity.

ACKNOWLEDGEMENTS

This work was supported by the National Key Research and Development Program of China (No. 2021ZD0113304), the National Science Foundation for Distinguished Young Scholars (with No. 62125604), and the NSFC projects (Key project with No. 61936010).

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

# A DERIVATIONS

## A.1 A PERSPECTIVE OF MINILLM FROM INVERSE REINFORCEMENT LEARNING

In Section 2.1, we formulate KD as an optimization problem of minimizing the discrepancy between the teacher distribution and the student distribution and finally reach the objective of minimizing *reverse* KLD. Alternatively, we can also regard KD as training the student model with the teacher model's guidance, which resembles an agent learning from the feedback from an environment. Following Pang & He (2021), we treat token generation as a Markov Decision Process. At each time step $t$, the student model chooses an action (token) $y_t$ from the action space (vocabulary) $V$ conditioning on the state (prefix) $(\boldsymbol{y}_{<t}, \boldsymbol{x})$ based on the policy (generation probability) $q_\theta(y_t|\boldsymbol{y}_{<t}, \boldsymbol{x})$.

From this perspective, standard KD corresponds to behavior cloning (BC; Torabi et al., 2018) in imitation learning (Ciosek, 2021). However, BC is known to under-perform Inverse Reinforcement Learning (IRL; Ziebart et al., 2008), another imitation learning method that first recovers a reward model from the environment demonstrations and then trains the policy with the reward using policy optimization algorithms (Sutton et al., 1999; Schulman et al., 2017). Therefore, in the KD scenario, we seek to first induce a reward $r(y_t, (\boldsymbol{y}_{<t}, \boldsymbol{x}))$ from the environment (the teacher model) and then train the student model to maximize the reward as the objective. We take the maximum-entropy inverse reinforcement learning framework (Ziebart et al., 2008; Chan & van der Schaar, 2021) and thus the Q-function $Q(y_t, (\boldsymbol{y}_{<t}, \boldsymbol{x}))$ in the environment satisfies the soft Bellman Equation:

$$Q(y_t, (\boldsymbol{y}_{<t}, \boldsymbol{x})) = r(y_t, (\boldsymbol{y}_{<t}, \boldsymbol{x})) + \gamma \log \sum_{y' \in V} \exp[Q(y', (\boldsymbol{y}_{\leq t}, \boldsymbol{x}))]. \tag{8}$$

We follow Hao et al. (2022) to parameterize the Q-function as $Q(y_t, (\boldsymbol{y}_{<t}, \boldsymbol{x})) = f(y_t, (\boldsymbol{y}_{<t}, \boldsymbol{x}))$ and assume $\gamma = 1$, where $f(y_t, (\boldsymbol{y}_{<t}, \boldsymbol{x}))$ is the output logits of the teacher model[4]. Then, the reward is given by:

$$r(y_t, (\boldsymbol{y}_{<t}, \boldsymbol{x})) = f(y_t, (\boldsymbol{y}_{<t}, \boldsymbol{x})) - \log \sum_{y' \in V} \exp[f(y', (\boldsymbol{y}_{\leq t}, \boldsymbol{x}))]. \tag{9}$$

To maximize the reward, we apply maximum-entropy reinforcement learning Haarnoja et al. (2017), whose learning objective is

$$\max_\theta \mathcal{J}(\theta) = \max_\theta \mathbb{E}_{\substack{\boldsymbol{x} \sim p_{\boldsymbol{x}} \\ \boldsymbol{y} \sim q_\theta(\cdot|\boldsymbol{x})}} \sum_{t=1}^{|\boldsymbol{y}|} [r(y_t, (\boldsymbol{y}_{<t}, \boldsymbol{x})) + \mathrm{H}[q_\theta(\cdot|\boldsymbol{y}_{<t}, \boldsymbol{x})]], \tag{10}$$

where $\mathrm{H}[q_\theta(\cdot|\boldsymbol{y}_{<t}, \boldsymbol{x})] = -\mathbb{E}_{y_t \sim q_\theta(\cdot|\boldsymbol{y}_{<t}, \boldsymbol{x})} \log q_\theta(\cdot|\boldsymbol{y}_{<t}, \boldsymbol{x})$ is the entropy of the student model distribution at the time step $t$.

**Equivalence Between Objectives** We prove an approximate equivalence between Eq. 10 and Eq. 1. We first rewrite the summation of the reward $\sum_{t=1}^{|\boldsymbol{y}|} r(y_t, (\boldsymbol{y}_{<t}, \boldsymbol{x}))$ by the associative law:

$$\sum_{t=1}^{|\boldsymbol{y}|} r(y_t, (\boldsymbol{y}_{<t}, \boldsymbol{x})) = \sum_{t=1}^{|\boldsymbol{y}|} \left[ f(y_t, (\boldsymbol{y}_{<t}, \boldsymbol{x})) - \log \sum_{y' \in V} \exp[f(y', (\boldsymbol{y}_{\leq t}, \boldsymbol{x}))] \right] \tag{11}$$

$$= f(y_1, (\boldsymbol{y}_{<1}, \boldsymbol{x})) + \sum_{t=2}^{|\boldsymbol{y}|} \left[ f(y_t, (\boldsymbol{y}_{<t}, \boldsymbol{x})) - \log \sum_{y' \in V} \exp[f(y', (\boldsymbol{y}_{<t}, \boldsymbol{x}))] \right] \tag{12}$$

$$- \log \sum_{y' \in V} \exp[f(y', (\boldsymbol{y}_{\leq|\boldsymbol{y}|}, \boldsymbol{x}))] \tag{13}$$

$$\approx \sum_{t=1}^{|\boldsymbol{y}|} \left[ f(y_t, (\boldsymbol{y}_{<t}, \boldsymbol{x})) - \log \sum_{y' \in V} \exp[f(y', (\boldsymbol{y}_{<t}, \boldsymbol{x}))] \right] \tag{14}$$

$$= \sum_{t=1}^{|\boldsymbol{y}|} \log \frac{\exp(f(y_t, (\boldsymbol{y}_{<t}, \boldsymbol{x})))}{\sum_{y' \in V} \exp(f(y', (\boldsymbol{y}_{<t}, \boldsymbol{x})))} \tag{15}$$

$$= \sum_{t=1}^{|\boldsymbol{y}|} \log p(y_t|\boldsymbol{y}_{<t}, \boldsymbol{x}). \tag{16}$$

---

[4]The teacher model's distribution satisfies $p(y_t|\boldsymbol{y}_{<t}, \boldsymbol{x}) = \frac{\exp(f(y_t, (\boldsymbol{y}_{<t}, \boldsymbol{x})))}{\sum_{y' \in V} \exp(f(y', (\boldsymbol{y}_{<t}, \boldsymbol{x})))}$.

Then, $\mathcal{J}(\theta)$ can be approximately rewritten as:

$$\mathcal{J}(\theta) \approx \underset{\substack{\boldsymbol{x} \sim p_{\boldsymbol{x}} \\ \boldsymbol{y} \sim q_\theta(\cdot|\boldsymbol{x})}}{\mathbb{E}} \sum_{t=1}^{|\boldsymbol{y}|} \left[ \log p(y_t|\boldsymbol{y}_{<t}, \boldsymbol{x}) + \mathrm{H}\left[ q_\theta(\cdot|\boldsymbol{y}_{<t}, \boldsymbol{x}) \right] \right] \tag{17}$$

$$= \underset{\substack{\boldsymbol{x} \sim p_{\boldsymbol{x}} \\ \boldsymbol{y} \sim q_\theta(\cdot|\boldsymbol{x})}}{\mathbb{E}} \sum_{t=1}^{|\boldsymbol{y}|} \left[ \log p(y_t|\boldsymbol{y}_{<t}, \boldsymbol{x}) - \log\left[ q_\theta(\cdot|\boldsymbol{y}_{<t}, \boldsymbol{x}) \right] \right] \tag{18}$$

$$= -\mathrm{KL}(q_\theta||p) \tag{19}$$

$$= -\mathcal{L}(\theta). \tag{20}$$

Therefore, maximizing $\mathcal{J}(\theta)$ is approximately equivalent to minimizing $\mathcal{L}(\theta)$.

## A.2 DERIVATION OF EQUATION 2

We compute the gradient of $\mathcal{L}(\theta) = \mathrm{KL}[q_\theta||p]$ with respect to $\theta$ using the Policy Gradient Theorem (Sutton et al., 1999):

$$\nabla \mathcal{L}(\theta) = -\nabla \underset{\substack{\boldsymbol{x} \sim p_{\boldsymbol{x}} \\ \boldsymbol{y} \sim q_\theta(\cdot|\boldsymbol{x})}}{\mathbb{E}} \log \frac{p(\boldsymbol{y}|\boldsymbol{x})}{q_\theta(\boldsymbol{y}|\boldsymbol{x})} \tag{21}$$

$$= -\int \nabla \left[ q_\theta(\boldsymbol{y}|\boldsymbol{x}) \log \frac{p(\boldsymbol{y}|\boldsymbol{x})}{q_\theta(\boldsymbol{y}|\boldsymbol{x})} \right] \mathrm{d}\boldsymbol{y}\mathrm{d}\boldsymbol{x} \tag{22}$$

$$= -\int q_\theta(\boldsymbol{y}|\boldsymbol{x}) \nabla \log \frac{p(\boldsymbol{y}|\boldsymbol{x})}{q_\theta(\boldsymbol{y}|\boldsymbol{x})} \mathrm{d}\boldsymbol{y}\mathrm{d}\boldsymbol{x} - \int \log \frac{p(\boldsymbol{y}|\boldsymbol{x})}{q_\theta(\boldsymbol{y}|\boldsymbol{x})} \nabla q_\theta(\boldsymbol{y}|\boldsymbol{x})\mathrm{d}\boldsymbol{y}\mathrm{d}\boldsymbol{x} \tag{23}$$

$$= \int q_\theta(\boldsymbol{y}|\boldsymbol{x}) \nabla \log q_\theta(\boldsymbol{y}|\boldsymbol{x})\mathrm{d}\boldsymbol{y}\mathrm{d}\boldsymbol{x} - \int q_\theta(\boldsymbol{y}|\boldsymbol{x}) \log \frac{p(\boldsymbol{y}|\boldsymbol{x})}{q_\theta(\boldsymbol{y}|\boldsymbol{x})} \nabla \log q_\theta(\boldsymbol{y}|\boldsymbol{x})\mathrm{d}\boldsymbol{y}\mathrm{d}\boldsymbol{x} \tag{24}$$

$$= -\underset{\substack{\boldsymbol{x} \sim p_{\boldsymbol{x}} \\ \boldsymbol{y} \sim q_\theta(\cdot|\boldsymbol{x})}}{\mathbb{E}} (\log \frac{p(\boldsymbol{y}|\boldsymbol{x})}{q_\theta(\boldsymbol{y}|\boldsymbol{x})} - 1) \nabla \log q_\theta(\boldsymbol{y}|\boldsymbol{x}) \tag{25}$$

$$= -\underset{\substack{\boldsymbol{x} \sim p_{\boldsymbol{x}} \\ \boldsymbol{y} \sim q_\theta(\cdot|\boldsymbol{x})}}{\mathbb{E}} \sum_{t=1}^{T}(\sum_{t'=1}^{T} \log \frac{p(y_{t'}|\boldsymbol{y}_{<t'}, \boldsymbol{x})}{q_\theta(y_{t'}|\boldsymbol{y}_{<t'}, \boldsymbol{x})} - 1) \nabla \log q_\theta(y_t|\boldsymbol{y}_{<t}, \boldsymbol{x}) \tag{26}$$

$$= -\underset{\substack{\boldsymbol{x} \sim p_{\boldsymbol{x}} \\ \boldsymbol{y} \sim q_\theta(\cdot|\boldsymbol{x})}}{\mathbb{E}} \sum_{t=1}^{T}(\sum_{t'=t}^{T} \log \frac{p(y_{t'}|\boldsymbol{y}_{<t'}, \boldsymbol{x})}{q_\theta(y_{t'}|\boldsymbol{y}_{<t'}, \boldsymbol{x})} - 1) \nabla \log q_\theta(y_t|\boldsymbol{y}_{<t}, \boldsymbol{x}), \tag{27}$$

where Eq. 27 is based on the fact that $\log q_\theta(y_t|\boldsymbol{y}_{<t}, \boldsymbol{x})$ can only affect tokens at $\geq t$ positions in $\boldsymbol{y}$. By setting $R_t = \sum_{t'=t}^{T} \log \frac{p(y_{t'}|\boldsymbol{y}_{<t'},\boldsymbol{x})}{q_\theta(y_{t'}|\boldsymbol{y}_{<t'},\boldsymbol{x})}$, we obtain Eq. 2.

## A.3 DERIVATION OF EQUATION 3

To derive Eq. 3, we first denote:

$$(\nabla \mathcal{L})_{\text{Single}} = -\underset{\substack{\boldsymbol{x} \sim p_{\boldsymbol{x}} \\ \boldsymbol{y} \sim q_\theta(\cdot|\boldsymbol{x})}}{\mathbb{E}} \left[ \sum_{t=1}^{T} \nabla \underset{y_t \sim q_\theta(t)}{\mathbb{E}} [r_t] \right],$$

$$(\nabla \mathcal{L})_{\text{Long}} = -\underset{\substack{\boldsymbol{x} \sim p_{\boldsymbol{x}} \\ \boldsymbol{y} \sim q_\theta(\cdot|\boldsymbol{x})}}{\mathbb{E}} \sum_{t=1}^{T} R_{t+1} \nabla \log q_\theta(y_t|\boldsymbol{y}_{<t}, \boldsymbol{x}). \tag{28}$$

Then, we re-write $\nabla \mathcal{L}(\theta)$ as:

$$\nabla \mathcal{L}(\theta) = - \mathop{\mathbb{E}}_{\substack{\boldsymbol{x} \sim p_{\boldsymbol{x}} \\ \boldsymbol{y} \sim q_\theta(\cdot|\boldsymbol{x})}} \sum_{t=1}^{T} (R_t - 1) \nabla \log q_\theta(y_t|\boldsymbol{y}_{<t}, \boldsymbol{x}) \tag{29}$$

$$= - \mathop{\mathbb{E}}_{\substack{\boldsymbol{x} \sim p_{\boldsymbol{x}} \\ \boldsymbol{y} \sim q_\theta(\cdot|\boldsymbol{x})}} \sum_{t=1}^{T} R_{t+1} \nabla \log q_\theta(y_t|\boldsymbol{y}_{<t}, \boldsymbol{x}) \tag{30}$$

$$- \mathop{\mathbb{E}}_{\substack{\boldsymbol{x} \sim p_{\boldsymbol{x}} \\ \boldsymbol{y} \sim q_\theta(\cdot|\boldsymbol{x})}} \sum_{t=1}^{T} \left( \log \frac{p(y_t|\boldsymbol{y}_{<t}, \boldsymbol{x})}{q_\theta(y_t|\boldsymbol{y}_{<t}, \boldsymbol{x})} - 1 \right) \nabla \log q_\theta(y_t|\boldsymbol{y}_{<t}, \boldsymbol{x}) \tag{31}$$

$$= (\nabla \mathcal{L})_{\text{Long}} - \mathop{\mathbb{E}}_{\substack{\boldsymbol{x} \sim p_{\boldsymbol{x}} \\ \boldsymbol{y} \sim q_\theta(\cdot|\boldsymbol{x})}} \sum_{t=1}^{T} \mathop{\mathbb{E}}_{y_t \sim q_\theta(\cdot|\boldsymbol{y}_{<t}, \boldsymbol{x})} \left( \log \frac{p(y_t|\boldsymbol{y}_{<t}, \boldsymbol{x})}{q_\theta(y_t|\boldsymbol{y}_{<t}, \boldsymbol{x})} - 1 \right) \nabla \log q_\theta(y_t|\boldsymbol{y}_{<t}, \boldsymbol{x}) \tag{32}$$

$$= (\nabla \mathcal{L})_{\text{Long}} - \mathop{\mathbb{E}}_{\substack{\boldsymbol{x} \sim p_{\boldsymbol{x}} \\ \boldsymbol{y} \sim q_\theta(\cdot|\boldsymbol{x})}} \sum_{t=1}^{T} \nabla \mathop{\mathbb{E}}_{y_t \sim q_\theta(\cdot|\boldsymbol{y}_{<t}, \boldsymbol{x})} \left[ - \log \frac{q_\theta(y_t|\boldsymbol{y}_{<t}, \boldsymbol{x})}{p(y_t|\boldsymbol{y}_{<t}, \boldsymbol{x})} \right] \tag{33}$$

$$= (\nabla \mathcal{L})_{\text{Long}} - \mathop{\mathbb{E}}_{\substack{\boldsymbol{x} \sim p_{\boldsymbol{x}} \\ \boldsymbol{y} \sim q_\theta(\cdot|\boldsymbol{x})}} \left[ \sum_{t=1}^{T} \nabla \mathop{\mathbb{E}}_{y_t \sim q_\theta(t)} [r_t] \right] \tag{34}$$

$$= (\nabla \mathcal{L})_{\text{Long}} + (\nabla \mathcal{L})_{\text{Single}}, \tag{35}$$

where Eq. 33 uses the product rule of the gradient and $r_t = \log \frac{p(y_t|\boldsymbol{y}_{<t}, \boldsymbol{x})}{q_\theta(y_t|\boldsymbol{y}_{<t}, \boldsymbol{x})}$.

# B  ALGORITHM DETAILS

---
**Algorithm 1** MINILLM: Knowledge Distillation of LLMs
---
**Input:**  Conditional generation dataset $\mathcal{D}$ consisting of prompts and ground-truth responses
        Pre-training corpus $\mathcal{D}_{\text{PT}}$ consisting of long-document plain texts
        A teacher model with output distribution $p$
        An initial student model pre-trained on $\mathcal{D}_{\text{PT}}$, with the output distribution $q_{\theta_0}$
        Learning rate $\eta$;    Batch size $M$;    Clipping Threshold $\epsilon$
**Output:**  A student model with the output distribution $q_\theta$
  Fine-tune the student model from $\theta_0$ on $\mathcal{D}$ supervised by the ground truth responses and choose $\theta$ with the lowest validation loss.
  **repeat**
    Sample a mini-batch of prompts from $\mathcal{D}$ and collect responses from $\widetilde{p}$ to get $\mathcal{S} = \{(\boldsymbol{x}^m, \boldsymbol{y}^m)\}_{m=1}^{M}$
    Sample a mini-batch $\mathcal{D}'_{\text{PT}} = \{\boldsymbol{d}^m\}_{m=1}^{M}$ from $\mathcal{D}_{\text{PT}}$
    Compute $(\nabla \mathcal{L})_{\text{Single}} = -\frac{1}{M} \sum_{\boldsymbol{x}, \boldsymbol{y} \in \mathcal{S}} \sum_{t=1}^{T} w_t \nabla \sum_{y_t \in V} q_\theta(y_t|\boldsymbol{y}_{<t}, \boldsymbol{x}) \log \frac{p(y_t|\boldsymbol{y}_{<t}, \boldsymbol{x})}{q_\theta(y_t|\boldsymbol{y}_{<t}, \boldsymbol{x})}$     ▷ Eq. 5
    Compute $(\nabla \mathcal{L})_{\text{Long}}^{\text{Norm}} = -\frac{1}{|M|} \sum_{\boldsymbol{x}, \boldsymbol{y} \in \mathcal{S}} \sum_{t=1}^{T} R_{t+1}^{\text{Norm}} \nabla \min[\rho_t(\theta), \text{clip}(\rho_t(\theta), 1 - \epsilon, 1 + \epsilon)]$,
    where $\rho_t(\theta) = \frac{q_\theta(y_t|\boldsymbol{y}_{<t}, \boldsymbol{x})}{\widetilde{p}(y_t|\boldsymbol{y}_{<t}, \boldsymbol{x})}$     ▷ Eq. 5, Eq. 6
    Compute the gradient of the language modeling loss: $\nabla \mathcal{L}_{\text{PT}} = -\frac{1}{M} \sum_{\boldsymbol{d} \in D'_{\text{PT}}} \nabla \log q_\theta(\boldsymbol{d})$
    Update model parameters: $\theta \leftarrow \theta - \eta \left[ (\nabla \mathcal{L})_{\text{Single}} + (\nabla \mathcal{L})_{\text{Long}}^{\text{Norm}} + \nabla \mathcal{L}_{\text{PT}} \right]$
  **until** converge and **return** $q_\theta$
---

# C  EXPERIMENTAL DETAILS

## C.1  TRAINING DETAILS

**Baselines**  Our baselines include **SFT w/o KD**, **KD**, and **SeqKD**. For models with less than 1.3B parameters, we search for the learning rates in [5e-4, 1e-4, 5e-5], the batch sizes in [32, 64], and train these models for 20 epochs. For other models, we search for the learning rate in [5e-5, 1e-5, 5e-6], the batch sizes in [32, 64], and train these models for 10 epochs. For **KD**, we follow Song et al. (2020) to mix the distillation loss with the language modeling loss on the ground truth responses by a mixture rate of 0.5. The checkpoints of each baseline are selected by the Rouge-L (Lin, 2004) scores on the validation set because, as stated in previous works (Wang et al., 2022; Ouyang et al., 2022), we also find that Rouge-L is better correlated with human judgments.

Below is an instruction that describes a task.
Write a response that appropriately completes the request.

### Instruction:
{instruction}

### Input:
{input}

### Response:

Figure 9: The prompt wrapper for training and evaluation.

Below is an instruction that describes a task.
Write a response that appropriately completes the request.

### Instruction:
Determine the sentiment of the input sentence. Please respond as positive or negative.

### Input:
{sentence}

### Response:

Figure 10: Zero-shot text classification prompt for SST2.

**MINILLM**   As stated in Section 2.3, training of MINILLM has two phases which is similar to Reinforcement Learning from Human Feedback (RLHF; Ouyang et al., 2022).

- **Phase 1**: We fine-tune the student model on the instruction-response training set $\mathcal{D}$ to get a starting point for the subsequent MINILLM training. We fine-tune the model for 3 epochs using the best learning rate and batch size of the corresponding **SFT w/o KD** baselines. Note that different from the **SFT w/o KD** baseline, we select the checkpoint with the *lowest validation loss*, not the Rouge-L score in this phase.
- **Phase 2**: We continuously train the model from **Phase 1** as described in Algorithm B using a learning rate 5e-6, a mini-batch size $64$ in all cases. The training and validation set are same as in **Phase 1**. Similar to Ouyang et al. (2022), we collect 256 sentences at once and adopt 4 inner epochs when doing the policy optimization. The clipping rate $\epsilon$ is set to 0.2, and the max length of the model is 512. We use temperature = 1 when sampling from $q_\theta$. We train the model for 5000 steps and select the final checkpoint using the Rouge-L score on the validation set. Our experiments are based on the NVIDIA V100 32G GPUs. Distilling LLaMA-7B from LLaMA-13B takes less than 10 ours on 16 GPUs.

### C.2   AUTOMATIC EVALUATION DETAILS

During the evaluation, we sample the responses from each model using temperature = 1, a max-length limit of 512, and random seeds [10, 20, 30, 40, 50]. Similar to Taori et al. (2023), we adopt a prompt wrapper shown in Figure 9 to convert each instruction-response pair to a sentence. For the classification tasks in the "Calibration" paragraph of Section 3.3, we prompt the model to do zero-shot text classification with the prompt in Figure 10 and 11.

### C.3   HUMAN EVALUATION DETAILS

Following Peng et al. (2023), we use SelfInst (Wang et al., 2023) to do the human evaluation. We randomly sampled 50 prompts because we found that more prompts do not affect the results much. We

---

Below is an instruction that describes a task.
Write a response that appropriately completes the request.

### Instruction:
Read the input passage and answer the question: {question}? Your answer should be "Yes" or "No".

### Input:
{passage}

### Response:

---

Figure 11: Zero-shot text classification prompt for BoolQ.

---

Below is an instruction that describes a task, paired with an input that provides further context. Write a response that appropriately completes the request.

### Instruction:
Desk jobs require writing a lot of emails, so it isn't surprising we get tired of repeating ourselves. Come up with several synonyms for the given word.

### Input:
Sincerely

### Response:

##### Answer #1 #####
Fondly, affectionately, lovingly, tenderly, honestly, truly, faithfully, devotedly, passionately

##### Answer #2 #####
Faithfully, Gullibly, Humbly, Piously, Strangely, Weirdly, Yours truly

1: Answer #1 is better
2: Answer #2 is better
3: Tie
Your choice:

---

Figure 12: The prompt wrapper for training and evaluation.

ask the annotators to compare the responses generated by the baseline models with the MINILLM and decide which response is preferred or neither of them is significantly better. Note that which model the responses come from is invisible to the annotators. The interface presented to annotators is shown in Figure 12.

### C.4 DETAILS ABOUT GENERATION DIVERSITY METRICS

In Table 3, we report the distinct 4-grams (Dist-4) and the language modeling loss (Loss) on the test sets. More details about these two metrics are as follows:

- **"Dist-4"** is a fraction: $N/C$, where $N$ is the number of the distinct 4-grams in the generated responses and $C$ is the total number of 4-grams. It is a widely used metric to measure the generation diversity of a language model (Li et al., 2016). The ($N/C$)s on the Dolly test set across 5 random seeds are shown in Table 5. Table 3 reports the average values across the 5 seeds.

| Seed
Model | 10 | 20 | 30 | 40 | 50 |
|---|---|---|---|---|---|
| Teacher | 23562 / 23696 | 23653 / 23834 | 24306 / 24488 | 24207 / 24381 | 23803 / 23967 |
| KD | 25889 / 26064 | 24024 / 24197 | 25663 / 25843 | 25611 / 25763 | 26178 / 26339 |
| SeqKD | 25358 / 25519 | 25631 / 25822 | 26190 / 26370 | 25574 / 25748 | 26295 / 26522 |
| MINILLM | 24187 / 24458 | 25011 / 25272 | 25100 / 25436 | 24067 / 24312 | 25205 / 25519 |

Table 5: The $(N/C)$s, where $N$ is the number of the distinct 4-grams in the generated responses and $C$ is the total number of 4-grams. We report the numbers computed on the Dolly test set when evaluated with 5 random seeds [10, 20, 30, 40, 50].

We would like to request your feedback on the performance of two AI assistants in response to the user instruction and input displayed above.

Please rate the helpfulness, relevance, accuracy, and level of detail of their responses. Each assistant receives an overall score on a scale of 1 to 10, where a higher score indicates better overall performance.

Please first output a single line containing only two values indicating the scores for Assistant 1 and 2, respectively. The two scores are separated by a space.

In the subsequent line, please provide a comprehensive explanation of your evaluation, avoiding any potential bias and ensuring that the order in which the responses were presented does not affect your judgment.

Figure 13: GPT-4 evaluation prompt.

- **"Loss"** is the negative log-likelihood loss on the test set $\mathcal{D}_{\text{Test}}$: $-\sum_{\boldsymbol{x},\boldsymbol{y}\sim\mathcal{D}_{\text{Test}}} \log q_\theta(\boldsymbol{y}|\boldsymbol{x})$. It measures the mode coverage of the real data distribution because it is essentially the forward KLD between the real data distribution and the model output distribution. This relates to diversity as in the ability to generate different generations given one context with different random seeds.

## C.5 EXPOSURE BIAS ANALYSIS

Following Arora et al. (2022), we compute the ExAccErr with the following formula:

$$\text{ExAccErr}(l) = \frac{R(l) - l\epsilon(l)}{l\epsilon(l)} \times 100\%, \tag{36}$$

where $R(l)$ is the accumulated regret of imitating the teacher distribution $p$ at the time step $l$ during the free-run generation:

$$R(l) = \sum_{t=1}^{T} \mathop{\mathbb{E}}_{\substack{\boldsymbol{y}_{<t}\sim q_\theta(\cdot|\boldsymbol{x}) \\ y_t\sim p(\cdot|\boldsymbol{y}_{<t},\boldsymbol{x})}} \log \frac{p(y_t|\boldsymbol{y}_{<t},\boldsymbol{x})}{q_\theta(y_t|\boldsymbol{y}_{<t},\boldsymbol{x})}, \tag{37}$$

and $\epsilon(l)$ is the average per-step error between $q_\theta$ and $p$ using the oracle context sampled from $p$ as the prefix:

$$\epsilon(l) = \frac{1}{l} \sum_{t=1}^{T} \mathop{\mathbb{E}}_{\substack{\boldsymbol{y}_{<t}\sim p(\cdot|\boldsymbol{x}) \\ y_t\sim p(\cdot|\boldsymbol{y}_{<t},\boldsymbol{x})}} \log \frac{p(y_t|\boldsymbol{y}_{<t},\boldsymbol{x})}{q_\theta(y_t|\boldsymbol{y}_{<t},\boldsymbol{x})}. \tag{38}$$

Intuitively, the regret of $q_\theta$ during generation is made of two parts: the error to estimate $p$ given the oracle context and the error caused by the low-quality model-generated prefix. The former is calculated by $l\epsilon(l)$, and the latter reflects the exposure bias. Therefore, ExAccErr measures the relative error caused only by exposure bias.

| Model | #Params | Method | Dolly | SelfInst | Vicuna |
|---|---|---|---|---|---|
| | 13B | Teacher | 79.0 | 75.5 | 65.1 |
| LLaMA | | SFT w/o KD | 73.0 | 69.2 | 61.6 |
| | 7B | KD | 73.7 | 70.5 | 62.7 |
| | | SeqKD | 73.6 | 71.5 | 62.6 |
| | | MINILLM | **76.4** | **73.1** | **64.1** |

Table 6: Evaluation results by GPT-4 feedback based on LLaMA family. We report the average GPT-4 feedback scores across 5 random seeds. The best scores of each model size are **boldfaced**.

| Model | Method | Dolly | SelfInst | Vicuna | S-NI | UnNI |
|---|---|---|---|---|---|---|
| GPT-J-6B | Teacher | 27.3 | 17.3 | 17.4 | 28.0 | 33.6 |
| GPT-2-760M | SFT w/o KD | 25.4 | 12.4 | 16.1 | 21.5 | 27.1 |
| | KD | **26.7** | 13.4 | 16.4 | 25.9 | 33.2 |
| | SeqKD | 26.0 | 14.0 | 15.3 | 25.5 | 32.5 |
| | MINILLM | 25.8 | **16.3** | **19.1*** | **27.1** | **35.5*** |
| GPT-2-1.5B | SFT w/o KD | **27.6*** | 14.3 | 16.3 | 27.6 | 34.6* |
| | KD | 26.6 | 14.5 | 16.5 | 27.6 | 34.9* |
| | SeqKD | 27.0 | 13.6 | 16.9 | 28.0 | 34.2* |
| | MINILLM | 25.9 | **16.6** | **19.4*** | **28.5*** | **35.9*** |
| GPT-*Neo*-2.7B | SFT w/o KD | 26.8 | 15.8 | 17.0 | 26.5 | 31.6 |
| | KD | 26.7 | 16.0 | 16.9 | 27.2 | 32.7 |
| | SeqKD | 25.6 | 16.2 | 16.9 | 26.1 | 32.9 |
| | MINILLM | **28.5*** | **17.1** | **18.6*** | **29.8*** | **35.4*** |

Table 7: Evaluation results when GPT-J is the teacher model. We report the average R-L scores across 5 random seeds. The best scores of each model size are **boldfaced**, and the scores where the student model outperforms the teacher are marked with *.

# D ADDITIONAL RESULTS

## D.1 GPT-4 FEEDBACK SCORES

We include the GPT-4 feedback scores (Zheng et al., 2023) on the LLaMA family as a supplementary evaluation metric, by asking GPT-4 to compare model-generated responses with the ground truth answers[5] and raise 1-10 scores for both responses. We apply the prompt in Figure 13 and set the temperature = 0.7 to call the GPT-4 API[6]. We report the ratio of the total score of model responses and ground truth answers. This metric is only applied to Dolly, SelfInst, and Vicuna. The results in Table 6 indicate that MINILLM has the best overall performance.

## D.2 GPT-J AS THE TEACHER MODEL

We present the evaluation results when using GPT-J as the teacher and GPT-2-760M, GPT-2-1.5B, and GPT-*Neo*-2.7B (Black et al., 2021) as the student in Table 7. MINILLM outperforms the baselines in most cases.

## D.3 SCALING LAW OF TEACHER BASED ON THE OPT FAMILY

We present the performance of MINILLM and SeqKD when we scale up the sizes of the teacher models in Figure 14. Similar to the observations in Section 3.3, MINILLM constantly performs better and distills better student models from larger teacher models.

---

[5]We use the ChatGPT's generation (OpenAI, 2022) for Vicuna's ground truth responses.
[6]We use the API version of 2023-3-15.

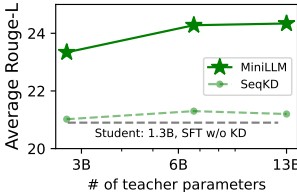

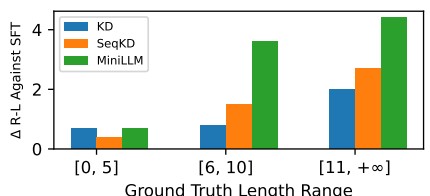

Figure 14: The scaling law of teacher based on the OPT family models. We compare MINILLM and SeqKD with OPT-1.3M as the student and OPT 2.7B, 6.7B, and 13B as teachers.

Figure 15: The Rouge-L scores of the distilled models against the SFT models on the different evaluation subsets of UnNI split by the golden responses' length.

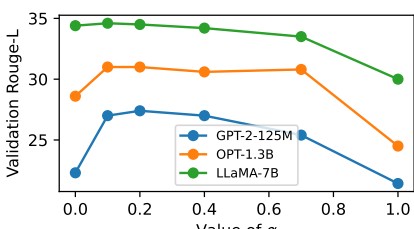

Figure 16: Effect of the $\alpha$ value in the teacher mix-in exploration on the validation Rouge-L score. Larger models to more robust to $\alpha$.

|      |                          | CLS  | Inst. |
|------|--------------------------|------|-------|
| 1.3B | MINILLM                  | **70.2** | **52.8** |
|      | w/o $\mathcal{L}_{PT}$   | 65.7 | 53.2  |
| 7B   | MINILLM                  | **78.8** | **71.2** |
|      | w/o $\mathcal{L}_{PT}$   | 74.3 | 71.1  |

Table 8: The effect of adding the pre-training loss. "CLS" is the average accuracy scores on SST2 and BoolQ. "Inst." is the average Rouge-L score on Dolly, SelfInst, and Vicuna.

### D.4    PERFORMANCE OF RESPONSE LENGTH ON U-NI

The performance on different U-NI subsets split by the length of the ground truth response is shown in Figure 15. We have the same observation as in Section 3.3 that on short responses, all KD methods perform similarly, and on long responses, MINILLM outperforms other methods.

### D.5    MORE ABLATION STUDIES

**Effect of Teacher-Mix-in Strength $\alpha$**    In Figure 16, we plot the best Rouge-L scores on the validation set of GPT-2-125M, OPT-1.3B, and LLaMA-7B using GPT-2-1.5B, OPT-13B, and LLAMA-13B as the teachers, with different teacher-mix-in strength $\alpha$ in MINILLM. $\alpha = 0.0$ means we only sample from the student distribution, and when $\alpha = 1.0$, we sample entirely from the teacher distribution. We find that $\alpha = 0.2$ is generally suitable across different model families and sizes, and larger models are more robust to the choice of $\alpha$.

**Effect of Adding Pre-Training Loss**    In Table 8, we study the effect of adding the pre-training loss in Algorithm B by comparing MINILLM with its variant where the language modeling loss on the pre-training corpus is removed (w/o $\mathcal{L}_{PT}$). We have a similar observation as Ouyang et al. (2022) that adding the pre-training loss helps to preserve the abilities on canonical NLP tasks while keeping the performance on instruction-following tasks nearly unchanged.

## E    CASES

We provide some cases generated by the models distilled by different methods based on the LLaMA model family in Table 9. The prompts are sampled from the SelfInst dataset. We find that MINILLM generates more detailed and accurate responses compared with the baselines.

| Case #1 | |
|---|---|
| Instruction | During writing, we added an asterisk for the word that did not come to mind. You will need to provide several examples to demonstrate all the words that can be used in the sentence instead of the asterisk. |
| Input | we * to know |
| SFT w/o KD | We can use "know" in the sentence instead of "we * to know" |
| KD | we went to know\n we will go to know\n we can go to know\n we may go to know\n we should go to know\n we must go to know |
| SeqKD | we do to know\n we can do to know\n we should do to know\n we would do to know\n we ought to do to know |
| MINILLM | We can use the following words in place of the asterisk:\n - want\n - have\n - do\n - used\n - ought\n - desire\n - need\n - must |
| MINILLM w/o teacher-mixed | We to know |
| Case #2 | |
| Instruction | Take the title of the product and extract its attributes. The attributes in this case refer to the characteristics of the products, such as their brand and color. Your answer must a list following the format of "attribute: value". |
| Input | SAMSUNG 980 PRO SSD 2TB PCIe NVMe Gen 4 Gaming M.2 Internal Solid State Hard Drive Memory Card, Maximum Speed, Thermal Control, MZ-V8P2T0B |
| SFT w/o KD | Attribute: Maximum Speed\n Value: 3500 MB/s |
| KD | attribute: brand, attribute: color, attribute: memory card, attribute: maximum speed, attribute: thermal control |
| SeqKD | brand: Samsung, color: black, size: 2TB, interface: PCIe NVME Gen 4, design: M.2 Internal |
| MINILLM | brand: SAMSUNG\n color: 980 PRO\n capacity: 2TB\n interface: PCIe NVME Gen 4\n speed: Maximum Speed\n threshold: Thermal Control\n form factor: M.2 Internal |
| MINILLM w/o teacher-mixed | SAMSUNGC 980 Pro SSD 2TB PCIe NVME Gen 4 Gaming M.II Internal Solid State Hard Drive memory Card, Max Speed, Thermal Control, AP1500K |

Table 9: Instruction-following cases from the SelfInst dataset. MINILLM better follows the instructions and provides more detailed and accurate responses. Without the teacher-mixed sampling strategy in Section 2.2, the distilled model outputs short responses (Case #1) or simply repeats the input (Cases #2).

