# OpenReview forum: "MiniLLM: Knowledge Distillation of Large Language Models"
_ICLR.cc/2024/Conference — ICLR 2024 poster_

### Official Review · Reviewer_oknc · 2023-10-21

**Soundness:** 2 fair
**Presentation:** 3 good
**Contribution:** 2 fair
**Rating:** 6
**Confidence:** 4

**Summary:**

For lowering computation cost required by large deep learning models, knowledge distillation (KD) is a popular approach. This study presents a knowledge distillation method applied to open-sourced large language models (LLMs). While the standard KD method uses a linear combination of cross entropy loss and Kullback-Leibler (KL) divergence referred to as forward KL divergence in this study, this study also examines a reverse KL divergence, which swaps teacher and student distributions and is used in computer vision and reinforcement learning literature. With the modified loss function, LMs trained on instruction-following datasets with teachers by their proposed approach (called MiniLLM) achieved higher average GPT-4 feedback scores than those trained with the same teacher model by a sequence-level KD (SeqKD) baseline.

**Strengths:**

- The reviewer wants to value originality of this study as this study is focused on open-sourced LLMs as targets for knowledge distillation, and black-box APIs can change their internal behavior without notice or proper versioning.
- This paper seems to well describe the proposed method and cites prior studies that inspired the authors to introduce key concepts in their method such as reverse KL divergence.
- It is empirically shown that MiniLLMs achieved better performance than KD baselines considered in this paper with multiple evaluation settings and instruction-following datasets. It is also notable that the overall trend in Table 1 seem consistent over different student models in a variety of model sizes.
- The ablation study attempts to test multiple hypotheses made when designing the proposed loss function.

**Weaknesses:**

## Presentation
This paper needs to improve the presentation and writing.
e.g.,

- MiniLLM model must be tautology (Mini large language model model) and should be referred to as just MiniLLM instead
- "distill <student model> from <teacher model>" sounds weird to the reviewer, and the reviewer suggests "distill (knowledge of) <teacher model> into <student model>"
- Itemized lists in this paper look very packed. Did the authors change the format and reduce space between items?
- "generative LLMs" and "generative language models" also sound strange as language models themselves are generative models. The reviewer suggests just skipping "generative".
- "the vocab size" should be "the vocabulary size"
- "similar to Learning from Human Feedback (RLFH; Ouyang et al., 2022)." misses "Reinforcement"
- Use [] for the second equation in Eq. (5) as well

## GPT-4-based evaluation

Overall experimental designs in this study look good, but the reviewer has a big concern about evaluations involving GPT-4. Specifically, it is very questionable how scientifically meaningful the evaluations are when leaving all the evaluations to GPT-4, and the reviewer did not find any reasonable justifications of using GPT-4 as part of the evaluation process. Rouge-L should be sufficient for Tables 1 and 4, and the reviewer strongly recommends use of Rouge-L instead of GPT-4 feedback (score) for Figures 1 and 5. The reviewer will improve rating if GPT-4-based evaluations are removed and replaced with Rouge-L.

**Questions:**

1. What is the definition of "white-box" KD/model in this paper? White-box in this paper sounds misleading.
2. Why is the specific $w_t$ between Eqs. (5) and (6) expected to reduce the variance of the estimator in Eq. (5)?
3. What is the definition of "Exposure Bias" ?(conceptual definition, not mathematical definition)
4. What is "the responses' distinct 4-gram proportion"?

---

> ### Author Response · Authors · 2023-11-18
> **Response to Reviewer oknc**
>
> We thank the reviewer for the thoughtful comments and suggestions.
>
> **Regarding the Presentation:**
>
> We will follow the reviewer’s suggestions on presentation to revise our paper. For the itemized list in the paper, we did not use the `\itemize` command but instead used `\bullet`. We have changed it to `\itemize` for a better presentation.
>
> **Regarding GPT-4 Evaluation:**
>
> We have replaced the GPT-4 score in Figure 1 and 5 with Rouge-L.
> We use GPT-4 for evaluation because it is studied by previous literature[1,2], where GPT-4 shows good alignment with human evaluation. This metric is also widely used in previous works to evaluate instruction-following[3,4,5].
>
> **Regarding the Questions:**
>
> 1. We follow the definition in [6]. Black-box KD refers to the scenario where only the teacher model's API (only the generated sentences, not including the probabilities) is available, and other cases are white-box KD. “White-box” means we need the model parameters to obtain the output probabilities to compute reverse KLD. We have clarified the description of “white-box KD” in the introduction of the revised paper.
> 2. $w_t$ does not reduce the estimator's variance in Eq. 5. It is the important sampling weight to ensure that the gradient estimation by sampling from the teacher-mixed distribution $\widetilde{p}$ equals the original estimator.
> 3. Exposure bias is a mismatch between MLE training and inference[7,8]. During training, the model predicts the next token conditioning on the prefix from the real data distribution. However, the prefix is generated from the model itself during inference. Therefore, the distribution of the prefixes seen during inference might differ greatly from those encountered during training, leading to a mismatch.
> 4. “Distinct 4-gram”[9] is a fraction: $N/C$, where $N$ is the number of the distinct 4-grams in the generated responses and $C$ is the total number of 4-grams. It is a widely used metric to measure the generation diversity of a language model.
>
> [1] G-EVAL: NLG Evaluation using GPT-4 with Better Human Alignment
>
> [2] Judging LLM-as-a-Judge with MT-Bench and Chatbot Arena. 2023. In NeurIPS.
>
> [3] Vicuna: An Open-Source Chatbot Impressing GPT-4 with 90\%* ChatGPT Quality.
>
> [4] Instruction Tuning with GPT-4.
>
> [5] LIMA: Less Is More for Alignment. 2023. In NeurIPS.
>
> [6] A Survey on Model Compression for Large Language Models.
>
> [7] Scheduled Sampling for Sequence Prediction with Recurrent Neural Networks. 2015. In NeurIPS.
>
> [8] Sequence Level Training with Recurrent Neural Networks. 2016. In ICLR.
>
> [9] A diversity-promoting objective function for neural conversation models. 2016. In NAACL.

---

> > ### Comment · Reviewer_oknc · 2023-11-18
> >
> > The reviewer checked responses from the authors. The reviewer wants to thank them for their clarifications and updates.
> >
> > > We have replaced the GPT-4 score in Figure 1 and 5 with Rouge-L. We use GPT-4 for evaluation because it is studied by previous literature[1,2], where GPT-4 shows good alignment with human evaluation. This metric is also widely used in previous works to evaluate instruction-following[3,4,5].
> >
> > It may be ok if the authors use the GPT-4 score (feedback) as a supplemental evaluation metric (i.e., moving them to appendix), but it is still not either convincing enough or clear how scientifically meaningful the evaluations are, for the following reasons.
> >
> > - Being used in a few previous studies does not explain why the metric is scientifically meaningful for this study. [3] is a blog post, [4] failed to justify why GPT-4 feedback is meaningful, [5] analyzes "Inter-Annotator Agreement", but the protocol is unclear. Moreover, their assessment is based on labeling which response was better, or whether neither response was significantly better than the other, which is different from the scheme used in this study (score 1 - 10)
> > - GPT-4 is a black-box service, and its behaviors may change over time. Also, the service is updated in an untrackable manner, thus it is very challenging to reproduce experimental results based on such services (including [1] and [2]).
> >
> > >  $w_t$ does not reduce the estimator's variance in Eq. 5. It is the important sampling weight to ensure that the gradient estimation by sampling from the teacher-mixed distribution equals the original estimator.
> >
> > The reviewer needs more clarifications for this point.
> > It looks like the revised manuscript still claims setting $w_t \approx$ ...  to reduce the variance of the estimator in Eq. 5 as follows
> >
> > >> Therefore, we approximately set $w_t \approx$ ... to reduce the variance of the estimator in Eq. 5
> >
> > from the manuscript.

---

> > > ### Author Response · Authors · 2023-11-19
> > > **Further Response to Reviewer oknc**
> > >
> > > We thank the reviewer for their comments. Further responses and clarifications are as follows:
> > >
> > > **Regarding GPT-4 scores**
> > >
> > > We understand the reviewer's concerns about the GPT-4 feedback evaluation. We have moved all GPT-4 scores in our paper to the Appendix.
> > >
> > > **Regarding $w_t$:**
> > >
> > > Our choice of $w_t \approx \frac{q_{\theta}(y_t | y_{<t}, x)}{\widetilde{p}(y_{t} | y_{<t}, x)}$ reduces variance because the original importance weight $w_t=\prod_{t'=1}^{t} \frac{q_\theta(y_{t'}|y_{<t'}, x)}{\widetilde{p}(y_{t'} | y_{<t'}, x)}$ requires multiplying per-token importance weight over multiple time steps, and thus the variance of each step will accumulate to the estimator. Therefore, we truncate the products to only consider the current time step $t$. This approximation is also used in prior works[1,2,3], where its effectiveness is empirically verified.
> > >
> > > [1] Text Generation by Learning from Demonstrations. 2021. In ICLR.
> > >
> > > [2] Offline Reinforcement Learning: Tutorial, Review, and Perspectives on Open Problems.
> > >
> > > [3] A deep reinforcement learning chatbot.

---

> > > > ### Comment · Reviewer_oknc · 2023-11-19
> > > >
> > > > The reviewer thanks the authors for the quick updates. The rating was improved, conditioned on the updates.
> > > >
> > > > The reviewer also wants to see the clarification of $w_t$ in the manuscript (footnote is fine if the space is limited)

---

### Official Review · Reviewer_BHa2 · 2023-10-29

**Soundness:** 4 excellent
**Presentation:** 4 excellent
**Contribution:** 4 excellent
**Rating:** 8
**Confidence:** 5

**Summary:**

This paper proposes to use Reverse KL Divergence for distilling large-language models. The paper starts from the Reverse KLD objective in section 2, describes the difference with forward KLD and its advantage that it is mode-seaking that is preferred when student has low capacity. Then in section 2.2 they review the challenges of optimizing for Reverse KLD and revisit ideas from prior work to improve it. To resolve challenges of reverse KLD, they propose 3 strategies to mitigate:
1) Decompose gradient into the gradient for single-step prediction and long sentence prediction
2) Prevent reward hacking by mixing the teacher/student distributions for sampling next token
3) Normalize the reward to prefer longer sequences.
They refer to Reverse KLD together with their strategies as MiniLLM.
In section 3, they evaluate the effectiveness of MiniLLM on instruction-following generation tasks. They use various teacher/student architectures including GPT-2, OPT, and LLaMA. They compare to baselines with and without knowledge distillation. Section 3.2 provides positive improvements using MiniLLM and section 3.3 provides analysis that shows the method scales well, gives well-calibrated models, and generates diverse outputs. Section 3.4 provides ablations on the three elements of MiniLLM.

**Strengths:**

- Significant gains and improvements compared with various baselines. The results show that improvements increase as the teacher gets bigger and all students at all parameter counts improve. So consistently large improvements.
- Figure 1: MiniLM is 5% better than SeqKD on GPT4 score.
- Table 1: MiniLM is up to 10% better than SFT w/o KD while KD is up to 5% better and SeqKD is up to 1% better than KD.
- Table 1: MiniLM is up to 8% better than the teacher while no other baseline surpasses the teacher.
- Comprehensive evaluations and ablations.

**Weaknesses:**

- The results seem to point that Reverse KLD might be harder to tune and requires all the strategies in MiniLM for performing better than baselines. This can be a challenge for reproduction and further research. I am specifically pointing to Table 4 and comparing to baseline numbers in Table 1: Why is MiniLM without either length-normalization (DollyEval GPT-4 22.4) or teacher-student distribution mixing (36.1) is significantly worse than comparable SeqKD (41.2) or KD (40.3) or SFT w/o KD (38.6) in Table 1? Does that mean Reverse KLD is generally harder to train without these strategies?
- Wall-clock time analysis of the method compared with baselines should be discussed. What is the training efficiency? How slow is the training with the MiniLM loss compared with SFT w/o KD, KD, and SeqKD? If the method is slower per iteration, what if baselines are trained for more iterations to match the wall-clock time of the method? Would they match the performance gains?

**Questions:**

- Page 2, introduction, line 11: How can one force q, the teacher, to do something? The teacher is not learnable. I’m assuming this is a typo and p/q_theta should be exchanged.
- Why is this work “white-box” KD? How is the method using the parameters of the teacher and not just the predictions of the teacher, p(y|x)?
- All experiments seem to be on instruction-following generation tasks. How would this distillation method perform for pretraining only? Can it help speed-up the pre-training of small models?
- Table 1: Can you provide examples of cases where the student is better than the teacher and provide a qualitative analysis of why? 8% improvement should show consistent improved behavior.
- Figure 5: Does MiniLM benefit more from scaling the teacher compared with SeqKD? Can you report the relative improvements of SeqKD and MiniLM as the teacher is scaled compared with a base teacher? If yes, it is useful for future scaling endeavors.
- Figure 8: y-axis says “Forward KLD” but the caption says “reverse KLD”, which value is plotted?
- Does the method have any hyperparameters specific to MiniLLM? For example, is there any thresholding of the ratio of q/q or p/q in Eq. 7 or epsilon in the denominator? If so, can you provide ablations?

Suggestions
- MiniLLM is a self-contradictory name as it expands to Mini Large Language Model. Please consider changing it, for example, to MiniLM.
- Figure 1: Please consider adding more description of sequence-level KD in the caption or the intro close to the reference to Figure 1. A reader not familiar with the literature does not learn about SeqKD until page 5.
- Eq 1 and A.1: For consistency it would help to use the KL with negative sign throughout (Eq. 1 is without negative sign but Eq. 8 is with negative sign). It would also help to highlight the difference in Eq 13 and 14 by color. It’s hard to notice the difference.
- Eq 7: It might help to use single/long gradients to simplify this equation and other predefined terms. This equation is not easily digestible as a summary equation.

Typos:
- Page 2: “... approximately minimizes the forward KLD” -> “minimize”
- Page 3: “... the quality of the each …” -> “the quality of each”

---

> ### Author Response · Authors · 2023-11-18
> **Response to Reviewer BHa2 (part 1)**
>
> We thank the reviewer for the detailed and encouraging comments. We will follow the suggestions about the paper writing and consider a better name for our method.
>
> **Regarding the Weakness:**
>
> **1. Strategies to train MiniLLM**
>
> Since longer sentences tend to have larger reverse KLD, length normalization prevents the model from outputting short and simple responses. And teacher-mixed sampling migrates the reward-hacking in policy optimization[1]. We provide all our codes in the supplementary material and will open-source all the codes, data, and model checkpoints for reproduction and further research.
>
> **2. Training efficiency**
>
> The training time with MiniLLM is generally less than 2 times of SFT and KD. Note that since we consider KD approaches on the downstream datasets (e.g., instruction-following) where the number of training instances is small, the total training time will not increase much with MiniLLM.
>
> We also evaluate the results if we train the student model with vanilla KD for the same amount of time as MiniLLM based on the GPT-2-base model (training for 40 epochs). The results show that the original training time (20 epochs) is enough for the model to converge, and training for more steps hurts the performance.
> | Method         | Valid Rouge-L | Test Rouge-L |
> | -------------- | ------------- | ------------ |
> | KD (20 epochs) | 25.3          | 22.8         |
> | KD (40 epochs) | 24.3          | 21.0         |
> | MiniLLM        | 27.4          | 24.6         |
>
> **Regarding the Questions:**
>
> 1. Yes, this is a typo. We have fixed it in the revision.
> 2. We follow the definition in [2]. Black-box KD refers to the scenario where only the teacher model's API (only the generated sentences, not including the probabilities) is available[3,4], and other cases are white-box KD. In our case, we need the model parameters to obtain p(y|x) to compute the reverse KLD. We have clarified the description of "white-box KD" in the introduction of the revised paper.
> 3. We did not conduct experiments during the pre-training phase and will leave this to future work.
>
> (continue in part 2)

---

> > ### Author Response · Authors · 2023-11-18
> > **Response to Reviewer BHa2 (part 2)**
> >
> > 4. Two examples from the Unnatural-Instructions test set on which the 760M MiniLLM outperforms the 1.5B teacher model are as follows:
> >
> >    **Example 1**
> >
> >    ```
> >    ## Instruction:
> >    You are given a sequence of steps which describe how to perform some task. Each step might reference information from a previous step. Your job is to re-order the steps so that they can be performed in order and make sense.
> >    ## Input:
> >    [' Mix together flour, sugar, eggs and butter', ' Preheat oven to 375 degrees', ' Roll out dough on floured surface'].
> >    ## Ground Truth:
> >    [' Preheat oven to 375 degrees', ' Mix together flour, sugar, eggs and butter', ' Roll out dough on floured surface']
> >    ## Teacher Model:
> >    First, preheat oven to 374 degrees. Then, mix together flour, sugar, eggs, butter, and strawberry jam. Once the oven reaches the desired temperature of 374 degrees, you can begin preparing the delicious strawberry jam bars. Have a try!
> >    ## MiniLLM:
> >    1. Preheat oven to 375 degrees
> >    2. Mix together flour, sugar, eggs and butter
> >    3. Roll out dough on floured surface
> >    ```
> >
> >    **Example 2:**
> >    ```
> >    ## Instruction:
> >    You will be given a paragraph of text, followed by a list of questions. For each question, you must determine the answer based on information in the paragraph.
> >    ## Input:
> >    Passage: The process of photosynthesis is used by plants to convert light energy into chemical energy that can be later used to fuel the plants' activities. This process involves two main steps: light absorption and carbon fixation. During light absorption,plant pigment molecules absorb photons and transfer their energy to chlorophyll molecules in the thylakoid membrane. In carbon fixation,the plant uses this excited state energy to convert CO2 gas into glucose sugar.
> >    Question 1: What are the two main steps of photosynthesis?
> >    Question 2: Where does light absorption take place?
> >    Question 3: What happens during carbon fixation?
> >    ## Ground Truth:
> >    1.The two main steps of photosynthesis are light absorption and carbon fixation.
> >    2. Light absorption takes place in the thylakoid membrane.
> >    3. During carbon fixation, the plant uses excited state energy to convert CO2 gas into glucose sugar.
> >    ## Teacher Model:
> >    Light absorption and carbon fixation. Light absorption takes place when chlorophyll molecules in plant cells capture photons from sunlight during the process of photosynthesis. This absorption of light energy is a crucial step in the conversion of solar energy into chemical energy…
> >    ## MiniLLM:
> >    Answer 1: Light absorption and carbon fixation.
> >    Answer 2: Light absorption takes place in the chlorophyll molecule of the plant.
> >    Answer 3: In carbon fixation, the plant uses this excited state of energy to convert CO2 into glucose.
> >    ```
> >    From the examples, we observe that MiniLLM follows the instructions better when generating long responses. The teacher model follows the instructions at first but then generates contents unrelated to the instructions. We attribute this phenomenon to exposure bias in the teacher model[5]. During training, the teacher model predicts the next word conditioning on the prefix from real data distribution, but during inference, the prefix is from the teacher model’s distribution, introducing a discrepancy. And this discrepancy accumulates during generation, causing the responses to finally deviate from the correct answer (exposure bias).
> >
> >    In contrast, MiniLLM sees self-generated responses during training (see Algorithm 1 in Appendix B), migrating the discrepancy. This effect is also observed in [6]. To summarize, MiniLLM with less exposure bias is possible to outperform the teacher model with more exposure bias.
> >
> > 5. We report the relative improvement when scaling up the teacher model compared to using the 340M teacher model. We find that MiniLLM benefits slightly better than SeqKD from scaling up the teacher model.
> >    | Teacher Model Size | 340M | 760M | 1.5B |
> >    | ------------------ | ---- | ---- | ---- |
> >    | SeqKD              | 0.0% | 0.3% | 1.0% |
> >    | MiniLLM            | 0.0% | 1.4% | 3.4% |
> >
> > 6. The value is Reverse KLD. We have fixed this typo in the revision.
> > 7. The only specific hyper-parameter of MiniLLM is the $\alpha$ in Eq. 4, and we provide an ablation study in Appendix D.4. We find that $\alpha=0.2$ is generally suitable for models across sizes and larger models are more robust to the choice of $\alpha$.
> >
> > [1] Defining and Characterizing Reward Hacking. 2022. In NeurIPS.
> >
> > [2] A Survey on Model Compression for Large Language Models.
> >
> > [3] Vicuna: An Open-Source Chatbot Impressing GPT-4 with 90\%* ChatGPT Quality.
> >
> > [4] Instruction Tuning with GPT-4.
> >
> > [5] Why Exposure Bias Matters: An Imitation Learning Perspective of Error Accumulation in Language Generation. In ACL findings.
> >
> > [6] Text generation by learning from demonstrations. 2021. In ICLR.

---

> > > ### Comment · Reviewer_BHa2 · 2023-11-22
> > > **I thank the the authors, my concerns are resolved**
> > >
> > > I thank the authors for their response and overall I see my concerns resolved and recommend this paper for acceptance.
> > >
> > > **W.1** Reverse KLD might be harder to tune… This might be a challenge for reproduction and further research.
> > >
> > > My concern is resolved by the fact that the method has only one hyperparameter and the ablations on that hyperparameter in Figure 16 shows relative consistently of the selected value 0.2. Moreover, the 3 proposed strategies are shown to be useful across all experiments.
> > >
> > > **W.2.1** Wall-clock time … should be discussed.
> > >
> > > The rebuttal provides the rough ratio of the wall-clock time for MiniLLM vs KD. I encourage the authors to add a table with the breakdown of the exact wall-clock time ratios for each parts of the MiniLLM rather than the overall rough estimate of “less than 2x”.
> > >
> > > **W.2.2** What if baselines are trained for more iterations to match the wall-clock time.
> > >
> > > The rebuttal partially resolves my concern by providing KD results of 2x epochs. I encourage authors to provide similar results for SeqKD.
> > >
> > > **Questions**
> > >
> > > I thank the authors for answering my questions including providing examples of MiniLLM outputs compared with the teacher. I agree with the authors’ analysis of the outputs.

---

### Official Review · Reviewer_HmVQ · 2023-10-30

**Soundness:** 3 good
**Presentation:** 4 excellent
**Contribution:** 2 fair
**Rating:** 5
**Confidence:** 3

**Summary:**

This work primarily concerns distillation of large language models (LLMs) into smaller, more portable versions. Compared to standard distillation approaches, the authors advocate the substitution of reverse KL instead of the more typical forward KL divergence objective. This incentivizes the model to pursue mode-seeking behavior, rather than coverage-seeking behavior, leading to more precise answers, with a lower probability of generating data outside of the teacher distribution. A policy gradient objective function is modified with single-step decomposition, teacher-mixed sampling, and length normalization to further improve optimization. Experiments with three families of LLMs (GPT-2, OPT, LLaMA) on a variety of benchmarks show that the proposed method (MiniLLM) indeed outperforms the baseline distillation approaches, across a range of student sizes.

**Strengths:**

## S1. Relevant topic
Given the recent advances and widespread usage of LLMs, ways of compressing models for faster and cheaper deployment can have significant real-world impacts in promoting wider spread usage of such models. The proposed approach shows promise being able to reduce model size without suffering as much performance decay as the baseline approach (Fig 1). This can have the effect of allowing more people to run LLMs, with the hardware they have available.

## S2. Experiments
1) Models: The experiments are on 3 model families (GPT-2, OPT, LLaMA), across a range of student model sizes (3 each for GPT-2 and OPT). This is a pretty good spread and helps give a sense of how well the proposed method generalizes, and the relation between student model size and performance.
2) Evaluation: Models are evaluated on DollyEval, SelfInst, VicunaEval, S-NI, and UnNI. Rouge-L, GPT4, and human evaluation are used as metrics. Results are reported on 5 generations from separate random seeds, which gives a better sense of the evaluation’s reliability.
3) Results: The main results suggest that MiniLLM indeed outperform the baselines, and occasionally even the teacher in certain cases. Taken at face value, this is pretty impressive, as some of the student models are considerably smaller (>10x for GPT=2, 10x for OPT). However, I have some concerns about the metrics and whether they’re capturing the full picture here (see W1.1).

## S3. Writing
I’ve listed a few miscellaneous corrections below, but overall, the writing is fairly clear and well-written.

**Weaknesses:**

## W1. Reverse KL vs Forward KL
One of the primary contributions of the paper is the substitution of reverse KL Divergence for forward KL divergence. This leads the student model to pursue “mode-seeking behavior”, as opposed to “coverage-seeking behavior”. While this does cut down on unrealistic generation samples, the trade-off is that such an approach will cause much of the long tail to be lost as well. This leads to a couple concerns:
1) The loss of sample diversity is not captured by the paper’s metrics, which primarily focus on realism/precision, so the baseline forward KL divergence is at a distinct disadvantage. Specifically, the metrics specifically measure where forward KL is weakest (correctness of samples), while ignoring where it is strongest (sample coverage). As such, the evaluation is somewhat unfair. Some sort of metric that captures sample diversity may yield a different story.
2) The long-tail knowledge of LLMs is arguably one of their most impressive and valuable properties, so sacrificing this in the name of realism is somewhat disappointing. In particular, this also raises potential ethical or fairness issues, as loss of diversity could lead to loss of minority representation or amplification of stereotypes.
3) Why do we have to make this tradeoff in the first place? Why not use both forward and reverse KL (see Q1), as in [a]?

## W2. Novelty/Contributions
From the abstract and introduction, it would seem that that the primary contributions are a) the focus on white-box KD for generative LLMs and b) the substitution of reverse KL instead of the more typical forward KL. Additionally, an amalgam of modifications (single-step decomposition, teacher-mixed sampling, length normalization) improves the policy gradient objective function to the final form (Equation 7) used in this paper. Some concerns/questions:
1) This is not a reason for rejection in and of itself, but individually, neither of these are particularly new concepts. White-box KL has been explored in the past, and the merits of forward vs reverse KL (as well as other divergences) for generative modeling has also been well explored.
2) It’s not clear how related whitebox KD for generative LLMs and a reverse KL optimization are. In fact, they seem almost entirely orthogonal from each other.
3) It’s not clear from the Methods section how whitebox KL was used in the method. Where does having the teacher model’s parameters play a role?
4) The methods in Section 2.2 seems to be a series of cobbled together heuristics, and it appears that they aren’t necessarily novel either, as there are clear connections with (cited) prior work. I’m not necessarily saying that that’s a bad thing to have as part of the method, but it doesn’t appear to be something that should be counted as a contribution of this work.


## Miscellaneous:
- The name “MiniLLM” doesn’t fully capture the method. Yes, the model is a smaller LLM, but the same can be said of a distilled LLM learned by forward KL divergence as well, so the name fails to distinguish one of the primary points of the paper.
- pg 2: “finite-number classes” => “a finite number of classes”
- Appendix entries out of order
- pg 5: “Ouyang et al. (2022)” citation at top of the page should be parenthetical?
- pg 9: Sec 4 – Knowledge Distillation: [b] may be a relevant related work
- pg 9: Fig 8: The y-axis says “Forward KLD”, while the caption says “reverse KLD”. Also, doesn’t this graph imply that w/o teacher-mixed sampling is better, if trained long enough?

[a] Chen, Liqun, et al. "Symmetric variational autoencoder and connections to adversarial learning." AISTATS, 2018.\
[b] Liang, K., et al. "Mixkd: Towards efficient distillation of large-scale language models." ICLR 2021.

**Questions:**

Q1. Given that forward and reverse KL each clearly have their own advantages and disadvantages, why not use both, e.g. as in [a]? See also [c] for a more general treatment of f-divergences for generative modeling.

Q2. The student models outperforming the teacher model in Table 1 is somewhat surprising. Why do you think this is the case? Does it have to do with the specific choice of metrics? I’m somewhat doubtful for example that a 120M parameter GPT-2 model is truly outperforming the 1.5 B parameter teacher model.

[a] Chen, Liqun, et al. "Symmetric variational autoencoder and connections to adversarial learning." AISTATS, 2018.\
[c] Nowozin, Sebastian, Botond Cseke, and Ryota Tomioka. "f-gan: Training generative neural samplers using variational divergence minimization." NeurIPS 2016.

---

> ### Author Response · Authors · 2023-11-18
> **Response to Reviewer HmVQ (part 1)**
>
> We thank the reviewer for the detailed and thoughtful comments. We will follow the suggestions about the paper writing and consider a better name for our method.
>
> **Regarding W1 and Q1 (Forward KLD v.s. Reverse KLD):**
>
> **1. Diversity measurement**
>
> In Table 3, we report the fraction of responses’ distinct 4-grams and the test loss, which measures the diversity and mode coverage of the model generation. More discussion can be found in the “Generation Diversity” paragraph in Section 3.3. In summary, the diversity does not decrease much (Dist-4: 99.5 v.s. 99.0), and the student model still covers most of the modes of the teacher model (test loss: 3.89 v.s. 3.95).
>
> **2. The long tail part of the distribution**
>
> We argue that our method will not sacrifice the long-tail knowledge of LLMs but only cause the sampled sentences to be less diverse (for example, the model will output similar responses with different random seeds). This is because:
>
> + The long-tail knowledge of LLMs is reflected by the long-tail part of the joint distribution of the input $x$ and the output $y$: $p(x, y)$, but our method computes reverse KLD between the conditional distribution $p(y|x)$. For example, even if $p(x,y)$ is a long-tail part, $p(y|x)$ will still be high if $y$ is a suitable continuation of $x$ and be covered by the student model using reverse KLD as the KD objective. In addition, reverse KLD encourages the model to ignore the low-probability regions of $p(y|x)$, and these regions have been proven harmful to language generation in previous literature[1,2].
>
> + During MiniLLM training, we add a language modeling loss on the models’ pre-training corpus (equivalent to minimizing the forward KLD between the student output distribution and the human distribution) as a regularization to preserve the knowledge learned during pre-training.
> For potential fairness and ethical issues, we compute the HONEST score[3] on the provided dataset in [3] to measure the hurtful stereotypes of the llama-7B models’ completion. We find that the HONEST score is not affected much by the KD approaches (higher scores indicate more hurtful completions):
> | Model      | HONEST Score |
> | ---------- | ------------ |
> | SFT w/o KD | 14.86        |
> | KD         | 15.22        |
> | SeqKD      | 16.04        |
> | MiniLLM    | 15.06        |
>
> **3. Why reverse KLD**
>
> There is a trade-off between the mode-seeking and the coverage-seeking behavior because the small model has low capacity and cannot cover all the modes of the large model, as stated in Section 2. There are three reasons that we only use reverse KLD for KD on the instruction-following datasets in our experiments:
>
> + Inspired by [4], we add a language modeling loss $\mathcal{L}_\text{pt}$ on the plain-text corpus, which is already a regularization to avoid the student model losing much of the long tail.
>
> + We find that the linguistical diversity and the coverage of the real data distribution are not affected much by mainly considering reverse KLD (see the paragraph “Generation Diversity” in section 3.3).
> Besides the role of $\mathcal{L}_\text{pt}$, we also suspect that the student model has the capacity to cover the main modes of $p(y|x)$, and most long-tail parts of $p(y|x)$ are noises, as shown in previous works[1,2].
>
> + In our pilot experiments, we tried combining reverse and forward KLD by summing the losses with the ratio 1:1, based on GPT-2-base with GPT-2-xLarge as the teacher, and compared the Rouge-L scores on the test sets. The results are shown in the following table. We can see that the Rouge-L scores were not affected much, indicating that reverse KLD is enough for KD in our setting. As a result, to keep the simplicity of the method, we did not add the forward KLD in our setting.
>
>   | Model                | Dolly | SelfInst | Vicuna | S-NI | UnNI |
>   | -------------------- | ----- | -------- | ------ | ---- | ---- |
>   | MiniLLM              | 24.6  | 13.2     | 16.9   | 25.3 | 30.1 |
>   | MiniLLM+forvward KLD | 23.4  | 13.8     | 16.9   | 24.7 | 29.3 |
>
>   We believe there are other ways to combine reverse and forward KLD, like in [a] and [c], and will explore them in future work.
> (continue in part 2)

---

> > ### Author Response · Authors · 2023-11-18
> > **Response to Reviewer HmVQ (part 2)**
> >
> > **Regarding W2:**
> >
> > **Point 1 & 4 (Novelty)**
> >
> > We study the **white-box KD of LLMs on text generation tasks**, but previous works on white-box KD focus on the image and text classification models, which differ largely from current LLMs. KD of LLMs has its own difficulty, and directly transferring previous white-box KD methods to LLMs does not work well (see the KD lines in our experiments). This is because LLMs generate answers auto-regressively, yielding a much larger label space than image and text classification tasks. Given the low capacity of the student model, there should be a trade-off between the mode-seeking and mode-average behavior. We point out this issue and provide a solution.
> >
> > For the same reason, even if the properties of reverse v.s. forward KLD have been explored, applying reverse KLD to the KD of LLMs is still non-trivial, as stated in Section 2.2, and thus the three strategies are needed. Some of the strategies are inspired by prior work, but they are themselves novel:
> > + Single-step Decomposition: Prior work only shows the importance of the first-step reward, but we propose to decompose it from the total gradient.
> >
> > + Teacher-mixed Sampling: The idea of mixing the teacher distribution is novel and the cited works are about analyzing the reward-hacking phenomenon, and the derivation of importance sampling.
> > + Length Regularization: This strategy is inspired by our pilot experiments and unrelated to prior works.
> >
> > In summary, the contributions of our work are
> > + Discussion on the difficulty of knowledge distillation of LLMs, which is practical but far from solved.
> > + A novel KD objective for LLMs and a stable training approach.
> > + Extensive experiments on diverse datasets and model scaling property, which is essential for the practical use of the method on real-world LLMs.
> >
> > **Point 2 & 3 (Definition of white-box KD)**
> >
> > We follow the definition in [5], where black-box KD refers to the scenario where only the API of the teacher model is available (only the generated sentences, not including the output distribution)[6,7], and other cases are white-box KD. MiniLLM belongs to the white-box KD because we need the model parameters to compute output distribution. In other words, we study KD when the teachers’ output distribution is available, which is related to using reverse KLD for optimization. We have clarified the description of “white-box KD” in the introduction of the revised paper.
> >
> > **Regarding Question #2 (The student model outperforming the teacher model):**
> >
> > We suspect the reason is related to exposure bias, i.e., the discrepancy between the MLE training and text generation phases. Since the teacher model is fine-tuned with MLE on the instruction following dataset, it suffers from exposure bias. However, MiniLLM is trained with policy optimization, which mitigates exposure bias in language generation, verified by both prior work[8] and our experiments in Figure 6. Therefore, a small model with less exposure bias (MiniLLM) is entirely possible to outperform a large model with more exposure bias. The metrics we use are the common metrics to evaluate instruction-following (Rouge-L, GPT-4 feedback, human evaluation)[9,10]. We leave the study of how the choice of metrics related to this phenomenon to future work.
> >
> > **Regarding “doesn’t this graph (Figure 8) imply that w/o teacher-mixed sampling is better, if trained long enough?”**
> >
> > MiniLLM w/o Teacher-Mixed Sampling shows the reward hacking phenomenon[11], where the model obtains high reward (low reverse KLD) but low scores on the final evaluation metrics (Rouge-L and GPT4 feedback in Table 4).
> >
> > [1] The Curious Case of Neural Text Degeneration. 2020. In ICLR.
> >
> > [2] Tailoring Language Generation Models under Total Variation Distance. 2021. In ICLR.
> >
> > [3] HONEST: Measuring Hurtful Sentence Completion in Language Models. 2022. In ACL.
> >
> > [4] Training language models to follow instructions with human feedback. 2022. In NeurIPS.
> >
> > [5] A Survey on Model Compression for Large Language Models.
> >
> > [6] Vicuna: An Open-Source Chatbot Impressing GPT-4 with 90\%* ChatGPT Quality.
> >
> > [7] Instruction Tuning with GPT-4.
> >
> > [8] Text generation by learning from demonstrations. 2021. In ICLR.
> >
> > [9] Super-NaturalInstructions: Generalization via Declarative Instructions on 1600+ Tasks. 2022. In EMNLP.
> >
> > [10] LIMA: Less Is More for Alignment. 2023. In NeurIPS.
> >
> > [11] Defining and Characterizing Reward Hacking. 2022. In NeurIPS.

---

> ### Comment · Reviewer_HmVQ · 2023-11-20
>
> I thank the authors for their thorough responses, which I’ve read along with the other reviews (and author responses).
>
> ### Novelty
> While I acknowledge the authors effort in making reverse KL work for distilling LLMs in their paper, I still believe the novelty here is being overstated. In particular, Section 2.1 is still written as if this work is the first ever to consider reverse KL for generative modeling, without citing many of the prior work, which while perhaps pre-dating the recent trend of LLMs, is still relevant.
>
> ### Diversity
> On diversity, could the authors explain more what the metrics for diversity measure?
> - Dist-4: My understanding is that this is a measure of the number of unique 4-grams, but reported as some sort of proportion. What is the numerator and denominator in this proportion?
> - Loss: It’s not clear to me how this is measure of diversity. Rather this would seem more to be a measure of how probable the generated text is from the training distribution.
>
> From the response, I’m also getting a mixed bag for interpreting the motivation and the results. It feels like the authors are simultaneously trying to argue for the value of reverse KL because it prioritizes coverage over covering modes (because low probability regions are harmful), and yet they are also arguing that coverage is in fact still intact?
>
> ### Whitebox
> Thank you for the clarification on the definition of white box here. There’s still potential for confusion here though, as the authors are using whitebox to refer to having the teacher model’s output probabilities; in many adversarial attack papers, having access to the model’s output is still considered an assumption of blackbox attacks.

---

> > ### Author Response · Authors · 2023-11-20
> > **Further Response to Reviewer oknc**
> >
> > We thank the reviewer for their comments. Further responses and clarifications are as follows:
> >
> > **Regarding Novelty**
> >
> > We have clarified that "we study the property of reverse KLD for KD of LLMs" in section 2.1 and added relevant citations.
> >
> > **Regarding Diversity**
> >
> > + “Dist-4”[9] is a fraction: $N/C$, where $N$ is the number of the distinct 4-grams in the generated responses and $C$ is the total number of 4-grams. It is a widely used metric to measure the generation diversity of a language model.
> > + Loss: The loss we reported in the original response and Table 3 is the language modeling loss: $\text{Loss} = -\sum_{y\sim p_{\text{train}}} \log q_\theta(y|x)$. Therefore, it measures "how probable the instances in the training distribution are from the model distribution", rather than "how probable the generated text is from the training distribution". We have clarified this in the latest revision of our paper.
> >
> >   In other words, loss measures the mode coverage of the real data distribution because it is essentially the forward KLD between the real data distribution and the model output distribution. This relates to diversity as in the ability to generate different generations given one context with different random seeds. Although this diversity is sacrificed to some extent (MiniLLM 3.95 v.s. SeqKD 3.91 in Table 3), we argue this is acceptable in practice because (1) the absolute value of the loss increase is small (2) many applications of test generation only require one but precise answer (like summarization, question answering, and coding).
> >
> > The main point of the “Forward KLD v.s. Reverse KLD” part in our response is that using reverse KLD for MiniLLM
> > (1) **will not** lose the long-tail knowledge of LLMs (because we consider $p(y|x)$, not $p(x,y)$ and add $\mathcal{L}_{\text{PT}}$);
> >
> > (2) **will not** introduce fairness and ethical issues (verified by the HONEST metrics);
> >
> > (3) **will probably** lose the diversity of the generated sentences (because  low-probability regions of $p(y|x)$ are not covered), but the diversity loss is not large (verified by the results in Table 3);
> >
> > (4) is sufficient for KD of LLMs and simply combining forward KLD does not yield better performance.
> >
> > In summary, we do not argue that coverage is in fact still intact (the Dist-4 indeed decreases and the loss indeed increases). We argue that using reverse KLD allows the MiniLLM to ignore modes that are small or even noisy low-probability regions of **$p(y|x)$** (not $p(x,y)$) but still covering **major** (not all) parts of the teacher distribution, leading to a slight and acceptable drop in diversity and improvement in generation quality. This is sufficient for KD of LLMs in text generation.
> >
> > **White-box**
> >
> > In the context of KD for LLMs, it is a common practice to include the scenario where the output distributions are available in white-box KD[1,2,3], mainly to distinguish from the methods that only use the APIs (output distributions are generally unavailable in model APIs).
> > If the reviewer finds it necessary, we can clarify the difference between the “division of white and black box” in KD for LLMs and adversarial attacks in the footnote to prevent misunderstanding from readers familiar with the adversarial attacks.
> >
> > [1] A Survey on Model Compression for Large Language Models.
> >
> > [2] Lion: Adversarial Distillation of Closed-Source Large Language Model.
> >
> > [3] LaMini-LM: A Diverse Herd of Distilled Models from Large-Scale Instructions.

---

> > > ### Comment · Reviewer_HmVQ · 2023-11-22
> > >
> > > Thank you for adding the clarifications to Section 2.1.
> > >
> > > Dist-4: Given the definition of this metric, I find it a little surprising that everything is 99+%. Can you also report $N$ and $C$ separately?
> > >
> > > I still find the statements in bullets 1-4 to be a little strong. The statements "will not lose the long-tail knowledge of LLMs" and "using reverse KLD allows the MiniLLM to ignore modes that are small or even noisy low-probability regions" are contradictory as the long-tail is by definition low-probability. While I admire the authors efforts to quantify these aspects, metrics only capture part of the story, and it's still extremely difficult to tell what's actually being lost here.
> > >
> > > I remain fairly on the fence, but I'm willing to raise my score to a 6.

---

> ### Author Response · Authors · 2023-11-23
> **About the Rating in the Official Review**
>
> We thank the reviewer for **the willingness to raise the score to 6**. We have reported $N$ and $C$ in the latest version of our paper.
>
> However, we notice that the score in the rating of the official review is still 5. Could you please **also change this rating to 6**?

---

### Official Review · Reviewer_v2PG · 2023-11-01

**Soundness:** 3 good
**Presentation:** 3 good
**Contribution:** 3 good
**Rating:** 6
**Confidence:** 4

**Summary:**

The paper proposes a method called MiniLLM for knowledge distillation of large language models (LLMs). The method focuses on distilling smaller language models from generative larger language models. It replaces the forward Kullback-Leibler divergence (KLD) objective in standard knowledge distillation approaches with reverse KLD, which is more suitable for generative language models.

**Strengths:**

- The paper introduces an application for knowledge distillation of generative language models.
- The proposed method is supported by well-structured experiments and evaluation on various datasets, demonstrating its effectiveness in generating more precise responses with higher overall quality, lower exposure bias, better calibration, and higher long-text generation performance.

**Weaknesses:**

(major) The novelty of this paper is limited. It is just a simple application of reverse KL Divergence to knowledge distillation. However, distill models with reverse KLD have been researched before. The claim in the abstract "how to effectively distill the knowledge … is still under-explored" is not convincing. For example [1]. More importantly, this paper is not cited by the authors.

(minor) In Table 1, the student model even outperforms the teacher model which lacks intuition. Although the authors attributed such results to the exposure bias issue of teacher-forcing, I doubt there is an overfitting problem with the experiments. Could the authors provide the hyper-parameters and the variance of each experiment?

[1] Self-Knowledge Distillation via Dropout

**Questions:**

See Weaknessed.

---

> ### Author Response · Authors · 2023-11-18
> **Response to Reviewer v2PG**
>
> We thank the reviewer for the thoughtful comments.
>
> **Regarding Weakness #1 (Novelty):**
>
> Our paper focuses on the KD of **LLMs on language generation tasks**, which is essentially different from the KD in [1] and under-explored in previous literature. In [1], reverse KLD-based KD is applied to image classification tasks, where the output space is a finite set of labels. However, the output space of LLMs on text generation tasks is much more complex, consisting of discrete token sequences with unlimited length, making it difficult and non-trivial to apply reverse KLD to the KD of LLMs.
>
> Specifically, in [1], the gradient of reverse KLD can be directly computed using the student model’s output distribution. In contrast, as stated in Section 2.2, computing the reverse KLD between the student and teacher language models requires sampling sentences from $q_\theta$ in an auto-regressive manner, and thus Policy Gradient (and the strategies we proposed) is needed to calculate the gradient of the objective. Therefore, we argue that it is novel to apply reverse KLD to KD of LLMs and design algorithms to stabilize the training process. Moreover, since [1] is related to our method, we have added it to the reference in the revision.
>
> **Regarding Weakness #2 (The student model outperforming the teacher model):**
>
> The hyper-parameters of each experiment are provided in Appendix C.1. We train 1.5B and 760M GPT-2 models under seeds [10, 20, 30, 40, 50] and report the average Rouge-L scores together with the standard deviations:
>
> | Model          | Dolly      | SelfInst   | Vicuna     | S-NI       | UnNI       |
> | -------------- | ---------- | ---------- | ---------- | ---------- | ---------- |
> | Teacher (1.5B) | 27.2 (0.3) | 14.9 (0.6) | 16.5 (0.1) | 27.8 (0.7) | 33.1 (1.0) |
> | MiniLLM (760M) | 26.3 (0.2) | 16.0 (0.6) | 18.1 (0.1) | 28.3 (0.6) | 36.4 (0.8) |
>
> Note that we train the models on one training set (Dolly) but observe that the student model outperforms the teacher model on the other 4 test sets. Therefore, we argue that the student models generalize well.
>
> As reported in previous works[2], replacing teacher-forcing with policy optimization can improve the quality of the generated texts by migrating exposure bias, which is orthogonal to the benefit of KD. Since the teacher model is trained in a teacher-forcing manner, the student model can outperform the teacher model by migrating this defect.
>
> [1] Self-Knowledge Distillation via Dropout.
>
> [2] Text generation by learning from demonstrations. 2021. In ICLR.

---

### Author Response · Authors · 2023-11-18
**General response to all reviewers**

We sincerely thank all the reviewers for their detailed comments and constructive suggestions, which surely help us strengthen our paper! It is encouraging that all reviewers appreciate our comprehensive experiments and strong empirical results on various evaluation sets. In addition, Reviewer HmVQ believed our proposed approach is promising in practical use, allowing more people to run LLMs, and reviewer BHa2 affirmed the usefulness of our method for future scaling endeavors. In the following, we will respond to each reviewer's comments and questions respectively. We have also updated our manuscript to incorporate the reviewers' suggestions. The major updates are highlighted in blue in our revised submission.

---

### Author Response · Authors · 2023-11-21
**General response to all reviewers**

Thanks again for reviewing our paper. We hope that we were able to address your concerns in our response. As the deadline is approaching, please let us know if you have any further questions before the reviewer-author discussion period ends. We are glad to address your further concerns.

---

### Meta-Review · Area_Chair_Ao5S · 2023-12-05

**Metareview:**

This paper proposes a distillation method for large language models based on optimizing the reverse KL and not the forward KL.

**Justification For Why Not Higher Score:**

The method requires several tricks to stabilize training.

**Justification For Why Not Lower Score:**

There was a consensus among reviewers that this paper should be accepted. In particular, as it seems to provide good empirical benefits compared to baselines in extensive experiments.

---

### Decision · Program_Chairs · 2024-01-16

Accept (poster)